# Efficiently Parameterized Neural Metriplectic Systems

## Abstract

Metriplectic systems are learned from data in a way that scales quadratically in both the size of the state and the rank of the metriplectic data. Besides being provably energy conserving and entropy stable, the proposed approach comes with approximation results demonstrating its ability to accurately learn metriplectic dynamics from data as well as an error estimate indicating its potential for generalization to unseen timescales when approximation error is low. Examples are provided which illustrate performance in the presence of both full state information as well as when entropic variables are unknown, confirming that the proposed approach exhibits superior accuracy and scalability without compromising on model expressivity.

## 1 Introduction

The theory of metriplectic, also called GENERIC, systems [1, 2] provides a principled formalism for encoding dissipative dynamics in terms of complete thermodynamical systems that conserve energy and produce entropy. By formally expressing the reversible and irreversible parts of state evolution with separate algebraic brackets, the metriplectic formalism provides a general mechanism for maintaining essential conservation laws while simultaneously respecting dissipative effects. Thermodynamic completeness implies that any dissipation is caught within a metriplectic system through the generation of entropy, allowing for a holistic treatment which has already found use in modeling fluids [3, 4], plasmas [5, 6], and kinetic theories [7, 8].

From a machine learning point of view, the formal separation of conservative and dissipative effects makes metriplectic systems highly appealing for the development of phenomenological models. Given data which is physics-constrained or exhibits some believed properties, a metriplectic system can be learned to exhibit the same properties with clearly identifiable conservative and dissipative parts. This allows for a more nuanced understanding of the governing dynamics via an evolution equation which reduces to an idealized Hamiltonian system as the dissipation is taken to zero. Moreover, elements in the kernel of the learned conservative part are immediately understood as Casimir invariants, which are special conservation laws inherent to the phase space of solutions, and are often useful for understanding and exerting control on low-dimensional structure in the system. On the other hand, the same benefit of metriplectic structure as a "direct sum" of reversible and irreversible parts makes it challenging to parameterize in an efficient way, since delicate degeneracy conditions must be enforced in the system for all time. In fact, there are no methods at present for learning general metriplectic systems which scale optimally with both dimension and the rank of metriplectic data—an issue which this work directly addresses.

Precisely, metriplectic dynamics on the finite or infinite dimensional phase space $\mathcal{P}$ are generated by a free energy function(al) $F : \mathcal{P} \to \mathbb{R}$, $F = E + S$ defined in terms of a pair $E, S : \mathcal{P} \to \mathbb{R}$ representing energy and entropy, respectively, along with two algebraic brackets $\{\cdot, \cdot\}, [\cdot, \cdot] : C^\infty(\mathcal{P}) \times C^\infty(\mathcal{P}) \to C^\infty(\mathcal{P})$ which are bilinear derivations on $C^\infty(\mathcal{P})$ with prescribed symmetries and degeneracies $\{S, \cdot\} = [E, \cdot] = 0$. Here $\{\cdot, \cdot\}$ is an antisymmetric Poisson bracket, which is a Lie algebra realization on functions, and $[\cdot, \cdot]$ is a degenerate metric bracket which is symmetric and positive

semi-definite. When $\mathcal{P} \subset \mathbb{R}^n$ for some $n > 0$, these brackets can be identified with symmetric matrix fields $\boldsymbol{L} : \mathcal{P} \to \mathrm{Skew}_n(\mathbb{R})$, $\boldsymbol{M} : \mathcal{P} \to \mathrm{Sym}_n(\mathbb{R})$ satisfying $\{F, G\} = \nabla F \cdot \boldsymbol{L} \nabla G$ and $[F, G] = \nabla F \cdot \boldsymbol{M} \nabla G$ for all functions $F, G \in C^\infty(\mathcal{P})$ and all states $\boldsymbol{x} \in \mathcal{P}$. Using the degeneracy conditions along with $\nabla \boldsymbol{x} = \boldsymbol{I}$ and abusing notation slightly then leads the standard equations for metriplectic dynamics,

$$\dot{\boldsymbol{x}} = \{\boldsymbol{x}, F\} + [\boldsymbol{x}, F] = \{\boldsymbol{x}, E\} + [\boldsymbol{x}, S] = \boldsymbol{L}\nabla E + \boldsymbol{M}\nabla S,$$

which are provably energy conserving and entropy producing. To see why this is the case, recall that $\boldsymbol{L}^\mathsf{T} = -\boldsymbol{L}$. It follows that the infinitesimal change in energy satisfies

$$\dot{E} = \dot{\boldsymbol{x}} \cdot \nabla E = \boldsymbol{L}\nabla E \cdot \nabla E + \boldsymbol{M}\nabla S \cdot \nabla E = -\boldsymbol{L}\nabla E \cdot \nabla E + \nabla S \cdot \boldsymbol{M}\nabla E = 0,$$

and therefore energy is conserved along the trajectory of $\boldsymbol{x}$. Similarly, the fact that $\boldsymbol{M}^\mathsf{T} = \boldsymbol{M}$ is positive semi-definite implies that

$$\dot{S} = \dot{\boldsymbol{x}} \cdot \nabla S = \boldsymbol{L}\nabla E \cdot \nabla S + \boldsymbol{M}\nabla S \cdot \nabla S = -\nabla E \cdot \boldsymbol{L}\nabla S + \boldsymbol{M}\nabla S \cdot \nabla S = |\nabla S|_{\boldsymbol{M}}^2 \geq 0,$$

so that entropy is nondecreasing along $\boldsymbol{x}$ as well. Geometrically, this means that the motion of a trajectory $\boldsymbol{x}$ is everywhere tangent to the level sets of energy and transverse to those of entropy, reflecting the fact that metriplectic dynamics are a combination of noncanonical Hamiltonian ($\boldsymbol{M} = \boldsymbol{0}$) and generalized gradient ($\boldsymbol{L} = \boldsymbol{0}$) motions. Note that these considerations also imply the Lyapunov stability of metriplectic trajectories, as can be seen by taking $E$ as a Lyapunov function. Importantly, this also implies that metriplectic trajectories which start in the (often compact) set $K = \{\boldsymbol{x} \,|\, E(\boldsymbol{x}) \leq E(\boldsymbol{x}_0)\}$ remain there for all time.

In phenomenological modeling, the entropy $S$ is typically chosen from Casimirs of the Poisson bracket generated by $\boldsymbol{L}$, i.e. those quantities $C \in C^\infty(\mathcal{P})$ such that $\boldsymbol{L}\nabla C = \boldsymbol{0}$. On the other hand, the method which will be presented here, termed neural metriplectic systems (NMS), allows for all of the metriplectic data $\boldsymbol{L}, \boldsymbol{M}, E, S$ to be approximated simultaneously, removing the need for Casimir invariants to be known or assumed ahead of time. The only restriction inherent to NMS is that the metriplectic system being approximated is nondegenerate (c.f. Definition 3.1), a mild condition meaning that the gradients of energy and entropy cannot vanish at any point $\boldsymbol{x} \in \mathcal{P}$ in the phase space. It will be shown that NMS alleviates known issues with previous methods for metriplectic learning, leading to easier training, superior parametric efficiency, and better generalization performance.

**Contributions.** The proposed NMS method for learning metriplectic models offers the following advantages over previous state-of-the-art: **(1)** It approximates arbitrary nondegenerate metriplectic dynamics with optimal quadratic scaling in both the problem dimension $n$ and the rank $r$ of the irreversible dynamics. **(2)** It produces realistic, thermodynamically consistent entropic dynamics from unobserved entropy data. **(3)** It admits universal approximation and error accumulation results given in Proposition 3.7 and Theorem 3.9. **(4)** It yields exact energy conservation and entropy stability by construction, allowing for effective generalization to unseen timescales.

## 2 Previous and Related Work

Previous attempts to learn metriplectic systems from data separate into "hard" and "soft" constrained methods. Hard constrained methods enforce metriplectic structure by construction, so that the defining properties of metriplecticity cannot be violated. Conversely, methods with soft constraints relax some aspects of metriplectic structure in order to produce a wider model class which is easier to parameterize. While hard constraints are the only way to truly guarantee appropriate generalization in the learned surrogate, the hope of soft constrained methods is that the resulting model is "close enough" to metriplectic that it will exhibit some of the favorable characteristics of metriplectic systems, such as energy and entropy stability. Some properties of the methods compared in this work are summarized in Table 1.

**Soft constrained methods.** Attempts to learn metriplectic systems using soft constraints rely on relaxing the degeneracy conditions $\boldsymbol{L}\nabla S = \boldsymbol{M}\nabla E = \boldsymbol{0}$. This is the approach taken in [9], termed SPNN, which learns an almost-metriplectic model parameterized with generic neural networks through a simple $L^2$ penalty term in the training loss, $\mathcal{L}_{\mathrm{pen}} = |\boldsymbol{L}\nabla E|^2 + |\boldsymbol{M}\nabla S|^2$. This widens the space of allowable network parameterizations for the approximate operators $\boldsymbol{L}, \boldsymbol{M}$. While

the resulting model violates the first and second laws of thermodynamics, the authors show that reasonable trajectories are still obtained, at least when applied within the range of timescales used for training. A similar approach is taken in [10], which targets larger problems and develops an almost-metriplectic model reduction strategy based on the same core idea.

**Hard constrained methods.** Perhaps the first example of learning metriplectic systems from data was given in [11] in the context of system identification. Here, training data is assumed to come from a finite element simulation, so that the discrete gradients of energy and entropy can be approximated as $\nabla E(\boldsymbol{x}) = \boldsymbol{A}\boldsymbol{x}, \nabla S(\boldsymbol{x}) = \boldsymbol{B}\boldsymbol{x}$. Assuming a fixed form for $\boldsymbol{L}$ produces a constrained learning problem for the constant matrices $\boldsymbol{M}, \boldsymbol{A}, \boldsymbol{B}$ which is solved to yield a provably metriplectic surrogate model. Similarly, the work [12] learns $\boldsymbol{M}, E$ given $\boldsymbol{L}, S$ by considering a fixed block-wise decoupled form which trivializes the degeneracy conditions, i.e. $\boldsymbol{L} = [\star\ \boldsymbol{0}; \boldsymbol{0}\ \boldsymbol{0}]$ and $\boldsymbol{M} = [\boldsymbol{0}\ \boldsymbol{0}; \boldsymbol{0}\ \star]$. This line of thought is continued in [13] and [14], both of which learn metriplectic systems with neural network parameterizations by assuming this decoupled block structure. A somewhat broader class of metriplectic systems are considered in [15] using tools from exterior calculus, with the goal of learning metriplectic dynamics on graph data. This leads to a structure-preserving network surrogate which scales linearly in the size of the graph domain, but also cannot express arbitrary metriplectic dynamics due to the specific choices of model form for $\boldsymbol{L}, \boldsymbol{M}$.

A particularly inspirational method for learning general metriplectic systems was given in [16] and termed GNODE, building on parameterizations of metriplectic operators developed in [17]. GNODE parameterizes learnable reversible and irreversible bracket generating matrices $\boldsymbol{L}, \boldsymbol{M}$ in terms of state-independent tensors $\boldsymbol{\xi} \in (\mathbb{R}^n)^{\otimes 3}$ and $\boldsymbol{\zeta} \in (\mathbb{R}^n)^{\otimes 4}$: for $1 \leq \alpha, \beta, \gamma, \mu, \nu \leq n$, the authors choose $L_{\alpha\beta}(\boldsymbol{x}) = \sum_\gamma \xi_{\alpha\beta\gamma}\partial^\gamma S$ and $M_{\alpha\beta}(\boldsymbol{x}) = \sum_{\mu,\nu} \zeta_{\alpha\mu,\beta\nu}\partial^\mu E \partial^\nu E$, where $\partial^\alpha F = \partial F/\partial x_\alpha$, $\boldsymbol{\xi}$ is totally antisymmetric, and $\boldsymbol{\zeta}$ is symmetric between the pairs $(\alpha, \mu)$ and $(\beta, \nu)$ but antisymmetric within each of these pairs. The key idea here is to exchange the problem of enforcing degeneracy conditions $\boldsymbol{L}\nabla E = \boldsymbol{M}\nabla S = \boldsymbol{0}$ in matrix fields $\boldsymbol{L}, \boldsymbol{M}$ with the problem of enforcing symmetry conditions in tensor fields $\boldsymbol{\xi}, \boldsymbol{\zeta}$, which is comparatively easier but comes at the expense of underdetermining the problem. In GNODE, enforcement of these symmetries is handled by a generic learnable 3-tensor $\tilde{\boldsymbol{\xi}} \in (\mathbb{R}^n)^{\otimes 3}$ along with learnable matrices $\boldsymbol{D} \in \mathrm{Sym}_r(\mathbb{R})$ and $\boldsymbol{\Lambda}^s \in \mathrm{Skew}_n(\mathbb{R})$ for $1 \leq s \leq r \leq n$, leading to the final parameterizations $\xi_{\alpha\beta\gamma} = \frac{1}{3!}\left(\tilde{\xi}_{\alpha\beta\gamma} - \tilde{\xi}_{\alpha\gamma\beta} + \tilde{\xi}_{\beta\gamma\alpha} - \tilde{\xi}_{\beta\alpha\gamma} + \tilde{\xi}_{\gamma\alpha\beta} - \tilde{\xi}_{\gamma\beta\alpha}\right)$ and $\zeta_{\alpha\mu,\beta\nu} = \sum_{s,t} \Lambda^s_{\alpha\mu}D_{st}\Lambda^t_{\beta\nu}$. Along with learnable energy and entropy functions $E, S$ parameterized by multi-layer perceptrons (MLPs), the data $\boldsymbol{L}, \boldsymbol{M}$ learned by GNODE guarantees metriplectic structure in the surrogate model and leads to successful learning of metriplectic systems in some simple cases of interest. However, note that this is a highly redundant parameterization requiring $\binom{n}{3} + r\binom{n}{2} + \binom{r+1}{2} + 2$ learnable scalar functions, which exhibits $\mathcal{O}(n^3 + rn^2)$ scaling in the problem size because of the necessity to compute and store $\binom{n}{3}$ entries of $\boldsymbol{\xi}$ and $r\binom{n}{2}$ entries of $\boldsymbol{\Lambda}$. Additionally, the assumption of state-independence in the bracket generating tensors $\boldsymbol{\xi}, \boldsymbol{\zeta}$ is somewhat restrictive, limiting the class of problems to which GNODE can be applied.

A related approach to learning metriplectic dynamics with hard constraints was seen in [18], which proposed a series of GFINN architectures depending on how much of the information $\boldsymbol{L}, \boldsymbol{M}, E, S$ is assumed to be known. In the case that $\boldsymbol{L}, \boldsymbol{M}$ are known, the GFINN energy and entropy are parameterized with scalar-valued functions $f \circ \boldsymbol{P}_{\ker \boldsymbol{A}}$ where $f : \mathbb{R}^n \to \mathbb{R}$ ($E$ or $S$) is learnable and $\boldsymbol{P}_{\ker \boldsymbol{A}} : \mathbb{R}^n \to \mathbb{R}^n$ is orthogonal projection onto the kernel of the (known) operator $\boldsymbol{A}$ ($\boldsymbol{L}$ or $\boldsymbol{M}$). It follows that the gradient $\nabla(f \circ \boldsymbol{P}_{\ker \boldsymbol{A}}) = \boldsymbol{P}_{\ker \boldsymbol{A}}\nabla f(P_{\ker \boldsymbol{A}})$ lies in the kernel of $\boldsymbol{A}$, so that the degeneracy conditions are guaranteed at the expense of constraining the model class of potential energies/entropies. Alternatively, in the case that all of $\boldsymbol{L}, \boldsymbol{M}, E, S$ are unknown, GFINNs use learnable scalar functions $f$ for $E, S$ parameterized by MLPs as well as two matrix fields $\boldsymbol{Q}^E, \boldsymbol{Q}^S \in \mathbb{R}^{r \times n}$ with rows given by $\boldsymbol{q}^f_s = \left(\boldsymbol{S}^f_s\nabla f\right)^\intercal$ for learnable skew-symmetric matrices $\boldsymbol{S}^f_s$, $1 \leq s \leq r$, $f = E, S$. Along with two triangular $(r \times r)$ matrix fields $\boldsymbol{T_L}, \boldsymbol{T_M}$, this yields the parameterizations $\boldsymbol{L}(\boldsymbol{x}) = \boldsymbol{Q}^S(\boldsymbol{x})^\intercal(\boldsymbol{T_L}(\boldsymbol{x})^\intercal - \boldsymbol{T_L}(\boldsymbol{x}))\boldsymbol{Q}^S(\boldsymbol{x})$ and $\boldsymbol{M}(\boldsymbol{x}) = \boldsymbol{Q}^E(\boldsymbol{x})^\intercal(\boldsymbol{T_M}(\boldsymbol{x})^\intercal\boldsymbol{T_M}(\boldsymbol{x}))\boldsymbol{Q}^E(\boldsymbol{x})$, which necessarily satisfy the symmetries and degeneracy conditions required for metriplectic structure. GFINNs are shown to both increase expressivity over the GNODE method as well as decrease redundancy, since the need for an explicit order-3 tensor field is removed and the reversible and irreversible brackets are allowed to depend explicitly on the state $\boldsymbol{x}$. However, GFINNs still exhibit cubic scaling through the requirement of $rn(n-1) + r^2 + 2 = \mathcal{O}(rn^2)$ learnable functions, which is well above the theoretical minimum required to express a general metriplectic system and leads to difficulties in training the resulting models.

**Model reduction.** Finally, it is worth mentioning the closely related line of work involving model reduction for metriplectic systems, which began with [19]. As remarked there, preserving metriplecticity in reduced-order models (ROMs) exhibits many challenges due to its delicate requirements on the kernels of the involved operators. There are also hard and soft constrained approaches: the already mentioned [10] aims to learn a close-to-metriplectic data-driven ROM by enforcing degeneracies by penalty, while [20] directly enforces metriplectic structure in projection-based ROMs using exterior algebraic factorizations. The parameterizations of metriplectic data presented here are related to those presented in [20], although NMS does not require access to nonzero components of $\nabla E, \nabla S$.

## 3 Formulation and Analysis

The proposed formulation of the metriplectic bracket-generating operators $\boldsymbol{L}, \boldsymbol{M}$ used by NMS is based on the idea of exploiting structure in the tensor fields $\boldsymbol{\xi}, \boldsymbol{\zeta}$ to reduce the necessary number of degrees of freedom. In particular, it will be shown that the degeneracy conditions $\boldsymbol{L}\nabla S = \boldsymbol{M}\nabla E = \boldsymbol{0}$ imply more than just symmetry constraints on these fields, and that taking these additional constraints into account allows for a more compact representation of metriplectic data. Following this, results are presented which show that the proposed formulation is universally approximating on nondegenerate systems (c.f. Definition 3.1) and admits a generalization error bound in time.

### 3.1 Exterior algebra

Developing these metriplectic expressions will require some basic facts from exterior algebra, of which more details can be found in, e.g., [21, Chapter 19]. The basic objects in the exterior algebra $\bigwedge V$ over the vector space $V$ are multivectors, which are formal linear combinations of totally antisymmetric tensors on $V$. More precisely, if $I(V)$ denotes the two-sided ideal of the free tensor algebra $T(V)$ generated by elements of the form $\boldsymbol{v} \otimes \boldsymbol{v}$ ($\boldsymbol{v} \in V$), then the exterior algebra is the quotient space $\bigwedge V \simeq T(V)/I(V)$ equipped with the antisymmetric wedge product operation $\wedge$. This graded algebra is equipped with natural projection operators $P^k : \bigwedge V \to \bigwedge^k V$ which map between the full exterior algebra and the $k^{\text{th}}$ exterior power of $V$, denoted $\bigwedge^k V$, whose elements are homogeneous $k$-vectors. More generally, given an $n$-manifold $M$ with tangent bundle $TM$, the exterior algebra $\bigwedge(TM)$ is the algebra of multivector fields whose fiber over $x \in M$ is given by $\bigwedge T_x M$.

For the present purposes, it will be useful to develop a correspondence between bivectors $\mathsf{B} \in \bigwedge^2(\mathbb{R}^n)$ and skew-symmetric matrices $\boldsymbol{B} \in \text{Skew}_n(\mathbb{R})$, which follows directly from Riesz representation in terms of the usual Euclidean dot product. More precisely, supposing that $\boldsymbol{e}_1, ..., \boldsymbol{e}_n$ are the standard basis vectors for $\mathbb{R}^n$, any bivector $\mathsf{B} \in \bigwedge^2 T\mathbb{R}^n$ can be represented as $\mathsf{B} = \sum_{i<j} B^{ij} \boldsymbol{e}_i \wedge \boldsymbol{e}_j$ where $B^{ij} \in \mathbb{R}$ denote the components of $\mathsf{B}$. Define a grade-lowering action of bivectors on vectors through right contraction (see e.g. Section 3.4 of [22]), expressed for any vector $\boldsymbol{v}$ and basis bivector $\boldsymbol{e}_i \wedge \boldsymbol{e}_j$ as $(\boldsymbol{e}_i \wedge \boldsymbol{e}_j) \cdot \boldsymbol{v} = (\boldsymbol{e}_j \cdot \boldsymbol{v})\boldsymbol{e}_i - (\boldsymbol{e}_i \cdot \boldsymbol{v})\boldsymbol{e}_j$. It follows that the action of $\mathsf{B}$ is equivalent to

$$\mathsf{B} \cdot \boldsymbol{v} = \sum_{i<j} B^{ij}((\boldsymbol{e}_j \cdot \boldsymbol{v})\boldsymbol{e}_i - (\boldsymbol{e}_i \cdot \boldsymbol{v})\boldsymbol{e}_j) = \sum_{i<j} B^{ij} v_j \boldsymbol{e}_i - \sum_{j<i} B^{ji} v_j \boldsymbol{e}_i = \sum_{i,j} B^{ij} v_j \boldsymbol{e}_i = \boldsymbol{B}\boldsymbol{v},$$

where $\boldsymbol{B}^{\mathsf{T}} = -\boldsymbol{B} \in \mathbb{R}^{n\times n}$ is a skew-symmetric matrix representing $\mathsf{B}$, and we have re-indexed under the second sum and applied that $B^{ij} = -B^{ji}$ for all $i, j$. Since the kernel of this action is the zero bivector, it is straightforward to check that this string of equalities defines an isomorphism $\mathcal{M} : \bigwedge^2 \mathbb{R}^n \to \text{Skew}_n(\mathbb{R})$ from the 2$^{\text{nd}}$ exterior power of $\mathbb{R}^n$ to the space of skew-symmetric $(n \times n)$-matrices over $\mathbb{R}$: in what follows, we will write $\boldsymbol{B} \simeq \mathsf{B}$ rather than $\boldsymbol{B} = \mathcal{M}(\mathsf{B})$ for notational convenience. Note that a correspondence in the more general case of bivector/matrix fields follows in the usual way via the fiber-wise extension of $\mathcal{M}$.

### 3.2 Learnable metriplectic operators

It is now possible to explain the proposed NMS formulation. First, note the following key definition which prevents the consideration of unphysical examples.

**Definition 3.1.** A metriplectic system on $K \subset \mathbb{R}^n$ generated by the data $\boldsymbol{L}, \boldsymbol{M}, E, S$ will be called *nondegenerate* provided $\nabla E, \nabla S \neq \boldsymbol{0}$ for all $\boldsymbol{x} \in K$.

With this, the NMS parameterizations for metriplectic operators are predicated on an algebraic result proven in Appendix A.

**Lemma 3.2.** *Let $K \subset \mathbb{R}^n$. For all $\boldsymbol{x} \in K$, the operator $\boldsymbol{L} : K \to \mathbb{R}^{n \times n}$ satisfies $\boldsymbol{L}^\intercal = -\boldsymbol{L}$ and $\boldsymbol{L}\nabla S = \boldsymbol{0}$ for some $S : K \to \mathbb{R}$, $\nabla S \neq \boldsymbol{0}$, provided there exists a non-unique bivector field $\mathsf{A} : U \to \bigwedge^2 \mathbb{R}^n$ and equivalent matrix field $\boldsymbol{A} \simeq \mathsf{A}$ such that*

$$\boldsymbol{L} \simeq \left( \mathsf{A} \wedge \frac{\nabla S}{|\nabla S|^2} \right) \cdot \nabla S = \mathsf{A} - \frac{1}{|\nabla S|^2} \boldsymbol{A}\nabla S \wedge \nabla S.$$

*Similarly, for all $\boldsymbol{x} \in K$ a positive semi-definite operator $\boldsymbol{M} : K \to \mathbb{R}^{n \times n}$ satisfies $\boldsymbol{M}^\intercal = \boldsymbol{M}$ and $\boldsymbol{M}\nabla E = \boldsymbol{0}$ for some $E : K \to \mathbb{R}$, $\nabla E \neq \boldsymbol{0}$, provided there exists a non-unique matrix-valued $\boldsymbol{B} : K \to \mathbb{R}^{n \times r}$ and symmetric matrix-valued $\boldsymbol{D} : K \to \mathbb{R}^{r \times r}$ such that $r \leq n$ and*

$$\boldsymbol{M} = \sum_{s,t} D_{st} \left( \boldsymbol{b}^s \wedge \frac{\nabla E}{|\nabla E|^2} \right) \cdot \nabla E \otimes \left( \boldsymbol{b}^t \wedge \frac{\nabla E}{|\nabla E|^2} \right) \cdot \nabla E$$

$$= \sum_{s,t} D_{st} \left( \boldsymbol{b}^s - \frac{\boldsymbol{b}^s \cdot \nabla E}{|\nabla E|^2} \nabla E \right) \left( \boldsymbol{b}^t - \frac{\boldsymbol{b}^t \cdot \nabla E}{|\nabla E|^2} \nabla E \right)^\intercal,$$

*where $\boldsymbol{b}^s$ denotes the $s^{\text{th}}$ column of $\boldsymbol{B}$. Moreover, using $\boldsymbol{P}_f^\perp = \left( \boldsymbol{I} - \frac{\nabla f \nabla f^\intercal}{|\nabla f|^2} \right)$ to denote the orthogonal projector onto $\operatorname{Span}(\nabla f)^\perp$, these parameterizations of $\boldsymbol{L}, \boldsymbol{M}$ are equivalent to the matricized expressions $\boldsymbol{L} = \boldsymbol{P}_S^\perp \boldsymbol{A} \boldsymbol{P}_S^\perp$ and $\boldsymbol{M} = \boldsymbol{P}_E^\perp \boldsymbol{B} \boldsymbol{D} \boldsymbol{B}^\intercal \boldsymbol{P}_E^\perp$.*

*Remark* 3.3. Observe that the projections appearing in these expressions are the minimum necessary for guaranteeing the symmetries and degeneracy conditions necessary for metriplectic structure. In particular, conjugation by $\boldsymbol{P}_f^\perp$ respects symmetry and ensures that both the input and output to the conjugated matrix field lie in $\operatorname{Span}(\nabla f)^\perp$.

Lemma 3.2 demonstrates specific parameterizations for $\boldsymbol{L}, \boldsymbol{M}$ that hold for any nondegenerate metriplectic data and are core to the NMS method for learning metriplectic dynamics. While generally underdetermined, these expressions are in a sense maximally specific given no additional information, since there is nothing available in the general metriplectic formalism to determine the matrix fields $\boldsymbol{A}, \boldsymbol{B} \boldsymbol{D} \boldsymbol{B}^\intercal$ on $\operatorname{Span}(\nabla S), \operatorname{Span}(\nabla E)$, respectively. The following Theorem, also proven in Appendix A, provides a rigorous correspondence between metriplectic systems and these particular parameterizations.

**Theorem 3.4.** *The data $\boldsymbol{L}, \boldsymbol{M}, E, S$ form a nondegenerate metriplectic system in the state variable $\boldsymbol{x} \in K$ if and only if there exist a skew-symmetric $\boldsymbol{A} : K \to \operatorname{Skew}_n(\mathbb{R})$, symmetric postive semidefinite $\boldsymbol{D} : K \to \operatorname{Sym}_r(\mathbb{R})$, and generic $\boldsymbol{B} : K \to \mathbb{R}^{n \times r}$ such that*

$$\dot{\boldsymbol{x}} = \boldsymbol{L}\nabla E + \boldsymbol{M}\nabla S = \boldsymbol{P}_S^\perp \boldsymbol{A} \boldsymbol{P}_S^\perp \nabla E + \boldsymbol{P}_E^\perp \boldsymbol{B} \boldsymbol{D} \boldsymbol{B}^\intercal \boldsymbol{P}_E^\perp \nabla S.$$

*Remark* 3.5. Note that the proposed parameterizations for $\boldsymbol{L}, \boldsymbol{M}$ are not one-to-one but properly contain the set of valid nondegenerate metriplectic systems in $E, S$, since the Jacobi identity on $\boldsymbol{L}$ necessary for a true Poisson manifold structure is not strictly enforced. For $1 \leq i, j, k \leq n$, the Jacobi identity is given in components as $\sum_\ell L_{i\ell}\partial^\ell L_{jk} + L_{j\ell}\partial^\ell L_{ki} + L_{k\ell}\partial^\ell L_{ij} = 0$. However, this requirement is not often enforced in algorithms for learning general metriplectic (or even symplectic) systems, since it is considered subordinate to energy conservation and it is well known that both qualities cannot hold simultaneously in general [23].

## 3.3 Specific parameterizations

Now that Theorem 3.4 has provided a model class which is rich enough to express any desired metriplectic system, it remains to discuss what NMS actually learns. First, note that it is unlikely to be the case that $E, S$ are known *a priori*, so it is beneficial to allow these functions to be learnable alongside the governing operators $\boldsymbol{L}, \boldsymbol{M}$. For simplicity, energy and entropy $E, S$ are parameterized as scalar-valued MLPs with $\tanh$ activation, although any desired architecture could be chosen for this task. The skew-symmetric matrix field $\boldsymbol{A} = \boldsymbol{A}_{\text{tri}} - \boldsymbol{A}_{\text{tri}}^\intercal$ used to build $\boldsymbol{L}$ is parameterized through its strictly lower-triangular part $\boldsymbol{A}_{\text{tri}}$ using a vector-valued MLP with output dimension $\binom{n}{2}$,

229   which guarantees that the mapping $\boldsymbol{A}_{\mathrm{tri}} \mapsto \boldsymbol{A}$ above is bijective. Similarly, the symmetric matrix
230   field $\boldsymbol{D} = \boldsymbol{K}_{\mathrm{chol}} \boldsymbol{K}_{\mathrm{chol}}^{\intercal}$ is parameterized through its lower-triangular Cholesky factor $\boldsymbol{K}_{\mathrm{chol}}$, which
231   is a vector-valued MLP with output dimension $\binom{r+1}{2}$. While this choice does not yield a bijective
232   mapping $\boldsymbol{K}_{\mathrm{chol}} \mapsto \boldsymbol{D}$ unless, e.g., $\boldsymbol{D}$ is assumed to be positive definite with diagonal entries of fixed
233   sign, this does not hinder the method in practice. In fact, $\boldsymbol{D}$ should not be positive definite in general,
234   as this is equivalent to claiming that $\boldsymbol{M}$ is positive definite on vectors tangent to the level sets of $E$.
235   Finally, the generic matrix field $\boldsymbol{B}$ is parameterized as a vector-valued MLP with output dimension
236   $nr$. Remarkably, the exterior algebraic expressions in Lemma 3.2 require less redundant operations
237   than the corresponding matricized expressions from Theorem 3.4, and therefore the expressions from
238   Lemma 3.2 are used when implementing NMS. Figure 1 summarizes this information.

239   *Remark* 3.6. With these choices, the NMS parameterization of metriplectic systems requires only
240   $(1/2)\big((n+r)^2 - (n-r)\big) + 2$ learnable scalar functions, in contrast to $\binom{n}{3} + r\binom{n}{2} + \binom{r+1}{2} + 2$ for
241   the GNODE approach in [16] and $rn(n-1) + r^2 + 2$ for the GFINN approach in [18]. In particular,
242   NMS is quadratic in both $n, r$ with no decrease in model expressivity, in contrast to the cubic scaling
243   of previous methods.

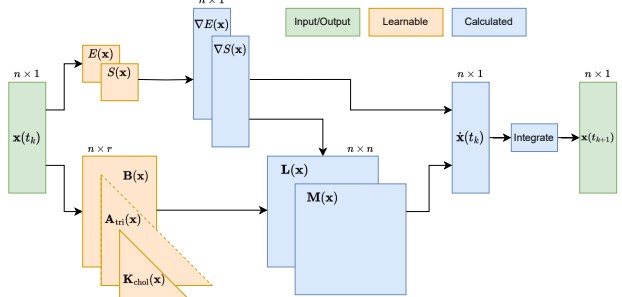

Table 1: Properties of the metriplectic architectures compared.

| Name | Physics Bias | Restrictive | Scale |
|------|------|------|------|
| NODE | None | No | Linear |
| SPNN | Soft | No | Quadratic |
| GNODE | Hard | Yes | Cubic |
| GFINN | Hard | No | Cubic |
| NMS | Hard | No | Quadratic |

Figure 1: A visual depiction of the NMS architecture.

## 3.4   Approximation and error

246   Besides offering a compact parameterization of metriplectic dynamics, the expressions used in NMS
247   also exhibit desirable approximation properties which guarantee a reasonable bound on state error
248   over time. To state this precisely, first note the following universal approximation result proven in
249   Appendix A.

250   **Proposition 3.7.** *Let $K \subset \mathbb{R}^n$ be compact and $E, S : K \to \mathbb{R}$ be continuous such that $\boldsymbol{L}\nabla S =$*
251   *$\boldsymbol{M}\nabla E = \boldsymbol{0}$ and $\nabla E, \nabla S \neq \boldsymbol{0}$ for all $\boldsymbol{x} \in K$. For any $\varepsilon > 0$, there exist two-layer neural network*
252   *functions $\tilde{E}, \tilde{S} : K \to \mathbb{R}, \tilde{\boldsymbol{L}} : K \to \mathrm{Skew}_n(\mathbb{R})$ and $\tilde{\boldsymbol{M}} : K \to \mathrm{Sym}_n(\mathbb{R})$ such that $\nabla \tilde{E}, \nabla \tilde{S} \neq \boldsymbol{0}$ on*
253   *$K$, $\tilde{\boldsymbol{M}}$ is positive semi-definite, $\tilde{\boldsymbol{L}}\nabla\tilde{S} = \tilde{\boldsymbol{M}}\nabla\tilde{E} = \boldsymbol{0}$ for all $\boldsymbol{x} \in K$, and each approximate function*
254   *is $\varepsilon$-close to its given counterpart on $K$. Moreover, if $\boldsymbol{L}, \boldsymbol{M}$ have $k \geq 0$ continuous derivatives on $K$*
255   *then so do $\tilde{\boldsymbol{L}}, \tilde{\boldsymbol{M}}$.*

256   *Remark* 3.8. The assumption $\boldsymbol{x} \in K$ of the state remaining in a compact set $V$ is not restrictive when
257   at least one of $E, -S : \mathbb{R}^n \to \mathbb{R}$, say $E$, has bounded sublevel sets. In this case, letting $\boldsymbol{x}_0 = \boldsymbol{x}(0)$ it
258   follows from $\dot{E} \leq 0$ that $E(\boldsymbol{x}(t)) = E(\boldsymbol{x}_0) + \int_0^t \dot{E}(\boldsymbol{x}(\tau))\, d\tau \leq E(\boldsymbol{x}_0)$, so that the entire trajectory
259   $\boldsymbol{x}(t)$ lies in the (closed and bounded) compact set $K = \{\boldsymbol{x} \mid E(\boldsymbol{x}) \leq E(\boldsymbol{x}_0)\} \subset \mathbb{R}^n$.

260   Leaning on Proposition 3.7 and classical universal approximation results in [24], it is further possible
261   to establish the following error estimate also proven in Appendix A which gives an idea of the error
262   accumulation rate that can be expected from this method.

263   **Theorem 3.9.** *Suppose $\boldsymbol{L}, \boldsymbol{M}, E, S$ are nondegenerate metriplectic data such that $\boldsymbol{L}, \boldsymbol{M}$ have at*
264   *least one continuous derivative, $E, S$ have Lipschitz continuous gradients, and at least one of $E, -S$*
265   *have bounded sublevel sets. For any $\varepsilon > 0$, there exist nondegenerate metriplectic data $\tilde{\boldsymbol{L}}, \tilde{\boldsymbol{M}}, \tilde{E}, \tilde{S}$*
266   *defined by two-layer neural networks such that, for all $T > 0$,*

$$\left( \int_0^T |\boldsymbol{x} - \tilde{\boldsymbol{x}}|^2 \, dt \right)^{\frac{1}{2}} \leq \varepsilon \left| \frac{b}{a} \right| \sqrt{e^{2aT} - 2e^{aT} + T + 1},$$

267 *where $a, b \in \mathbb{R}$ are constants depending on both sets of metriplectic data and $\dot{\tilde{\boldsymbol{x}}} = \tilde{\boldsymbol{L}}(\tilde{\boldsymbol{x}})\nabla\tilde{E}(\tilde{\boldsymbol{x}}) +$*
268 *$\tilde{\boldsymbol{M}}(\tilde{\boldsymbol{x}})\nabla\tilde{S}(\tilde{\boldsymbol{x}})$.*

269 *Remark* 3.10. Theorem 3.9 provides a bound on state error over time under the assumption that the
270 approximation error in the metriplectic networks can be controlled. On the other hand, notice that
271 Theorem 3.9 can also be understood as a generic error bound on any trained metriplectic networks
272 $\tilde{\boldsymbol{L}}, \tilde{\boldsymbol{M}}, \tilde{E}, \tilde{S}$ provided universal approximation results are not invoked in the estimation leading to $\varepsilon b$.

273 This result confirms that the error in the state $\boldsymbol{x}$ for a fixed final time $T$ tends to zero with the
274 approximation error in the networks $\tilde{\boldsymbol{L}}, \tilde{\boldsymbol{M}}, \tilde{E}, \tilde{S}$, as one would hope based on the approximation
275 capabilities of neural networks. More importantly, Theorem 3.9 also bounds the generalization error
276 of any trained metriplectic network for an appropriate (and possibly large) $\varepsilon$ equal to the maximum
277 approximation error on $K$, where the learned metriplectic trajectories are confined for all time.
278 With this theoretical guidance, the remaining goal of this work is to demonstrate that NMS is also
279 practically effective at learning metriplectic systems from data and exhibits reasonable generalization
280 to unseen timescales.

## 4 Algorithm

282 Similar to previous approaches in [16] and [18], the learnable parameters in NMS are calibrated
283 using data along solution trajectories to a given dynamical system. This brings up an important
284 question regarding how much information about the system in question is realistically present in
285 the training data. While many systems have a known metriplectic form, it is not always the case
286 that one will know metriplectic governing equations for a given set of training data. Therefore, two
287 approaches are considered in the experiments below corresponding to whether full or partial state
288 information is assumed available during NMS training. More precisely, the state $\boldsymbol{x} = (\boldsymbol{x}^o, \boldsymbol{x}^u)$ will
289 be partitioned into "observable" and "unobservable" variables, where $\boldsymbol{x}^u$ may be empty in the case
290 that full state information is available. In a partially observable system $\boldsymbol{x}^o$ typically contains positions
291 and momenta while $\boldsymbol{x}^u$ contains entropy or other configuration variables which are more difficult
292 to observe during physical experiments. In both cases, NMS will learn a metriplectic system in $\boldsymbol{x}$
293 according to the procedure described in Algorithm 1.

---

**Algorithm 1** Training neural metriplectic systems

---

1: **Input:** snapshot data $\boldsymbol{X} \in \mathbb{R}^{n \times n_s}$, each column $\boldsymbol{x}_s = \boldsymbol{x}(t_s, \boldsymbol{\mu}_s)$, target rank $r \geq 1$, batch size $n_b \geq 1$.
2: Initialize networks $\boldsymbol{A}_{\text{tri}}, \boldsymbol{B}, \boldsymbol{K}_{\text{chol}}, E, S$, and loss $L = 0$
3: **for** step in $N_{\text{steps}}$ **do**
4:     Randomly draw batch $P = \{(t_{s_k}, \boldsymbol{x}_{s_k})\}_{k=1}^{n_b}$
5:     **for** $(t, \boldsymbol{x})$ in $P$ **do**
6:         Evaluate $\boldsymbol{A}_{\text{tri}}(\boldsymbol{x}), \boldsymbol{B}(\boldsymbol{x}), \boldsymbol{K}_{\text{chol}}(\boldsymbol{x}), E(\boldsymbol{x}), S(\boldsymbol{x})$
7:         Automatically differentiate $E, S$ to obtain $\nabla E(\boldsymbol{x}), \nabla S(\boldsymbol{x})$
8:         Form $\boldsymbol{A}(\boldsymbol{x}) = \boldsymbol{A}_{\text{tri}}(\boldsymbol{x}) - \boldsymbol{A}_{\text{tri}}(\boldsymbol{x})^{\mathsf{T}}$ and $\boldsymbol{D}(\boldsymbol{x}) = \boldsymbol{K}_{\text{chol}}(\boldsymbol{x})\boldsymbol{K}_{\text{chol}}(\boldsymbol{x})^{\mathsf{T}}$
9:         Build $\boldsymbol{L}(\boldsymbol{x}), \boldsymbol{M}(\boldsymbol{x})$ according to Lemma 3.2
10:        Evaluate $\dot{\boldsymbol{x}} = \boldsymbol{L}(\boldsymbol{x})\nabla E(\boldsymbol{x}) + \boldsymbol{M}(\boldsymbol{x})\nabla S(\boldsymbol{x})$
11:        Randomly draw $n_1, ..., n_l$ and form $t_j = t + n_j \Delta t$ for all $j$
12:        $\tilde{\boldsymbol{x}}_1, ..., \tilde{\boldsymbol{x}}_l = \text{ODEsolve}(\dot{\boldsymbol{x}}, t_1, ..., t_l)$
13:        $L \mathrel{+}= l^{-1} \sum_j \text{Loss}(\boldsymbol{x}_j, \tilde{\boldsymbol{x}}_j)$
14:     **end for**
15:     Rescale $L = |P|^{-1} L$
16:     Update $\boldsymbol{A}_{\text{tri}}, \boldsymbol{B}, \boldsymbol{K}_{\text{chol}}, E, S$ through gradient descent on $L$.
17: **end for**

---

294 Note that the batch-wise training strategy in Algorithm 1 requires initial conditions for $\boldsymbol{x}^u$ in the
295 partially observed case. There are several options for this, and two specific strategies will be
296 considered here. Suppose the data are drawn from the training interval $[0, T]$ with initial state $\boldsymbol{x}_0$
297 and final state $\boldsymbol{x}_T$. The first strategy sets $\boldsymbol{x}_0^u = \boldsymbol{0}, \boldsymbol{x}_T^u = \boldsymbol{1}$ (where $\boldsymbol{1}$ is the all ones vector), and
298 $\boldsymbol{x}_s^u = \boldsymbol{1}/T, 0 < s < T$, so that the unobserved states are initially assumed to lie on a straight line.
299 The second strategy is more sophisticated, and involves training a diffusion model to predict the
300 distribution of $\boldsymbol{x}^u$ given $\boldsymbol{x}^o$. Specific details of this procedure are given in Appendix E.

# 5   Examples

The goal of the following experiments is to show that NMS is effective even when entropic information cannot be observed during training, yielding superior performance when compared to previous methods including GNODE, GFINN, and SPNN discussed in Section 2. The metrics considered for this purpose will be mean absolute error (MAE) and mean squared error (MSE) defined in the usual way as the average $\ell^1$ (resp. squared $\ell^2$) error between the discrete states $\boldsymbol{x}, \tilde{\boldsymbol{x}} \in \mathbb{R}^{n \times n_s}$. For brevity, many implementation details have been omitted here and can be found in Appendix B. An additional experiment showing the effectiveness of NMS in the presence of both full and partial state information can be found in Appendix C.

*Remark* 5.1. To facilitate a more equal parameter count between the compared metriplectic methods, the results of the experiments below were generated using the alternative parameterization $\boldsymbol{D} = \boldsymbol{K}\boldsymbol{K}^\intercal$ where $\boldsymbol{K} : K \to \mathbb{R}^{r \times r'}$ is full and $r' \geq r$. Of course, this change does not affect metriplecticity since $\boldsymbol{D}$ is still positive semi-definite for each $\boldsymbol{x} \in K$.

## 5.1   Two gas containers

The first test of NMS involves two ideal gas containers separated by movable wall which is removed at time $t_0$, allowing for the exchange of heat and volume. In this example, the motion of the state $\boldsymbol{x} = \begin{pmatrix} q & p & S_1 & S_2 \end{pmatrix}^\intercal$ is governed by the metriplectic equations:

$$\dot{q} = \frac{p}{m}, \qquad\qquad \dot{p} = \frac{2}{3}\left(\frac{E_1(\boldsymbol{x})}{q} - \frac{E_2(\boldsymbol{x})}{2L - q}\right),$$

$$\dot{S}_1 = \frac{9N^2 k_B^2 \alpha}{4E_1(\boldsymbol{x})}\left(\frac{1}{E_1(\boldsymbol{x})} - \frac{1}{E_2(\boldsymbol{x})}\right), \qquad \dot{S}_2 = -\frac{9N^2 k_B^2 \alpha}{4E_1(\boldsymbol{x})}\left(\frac{1}{E_1(\boldsymbol{x})} - \frac{1}{E_2(\boldsymbol{x})}\right),$$

where $(q, p)$ are the position and momentum of the separating wall, $S_1, S_2$ are the entropies of the two subsystems, and the internal energies $E_1, E_2$ are determined from the Sackur-Tetrode equation for ideal gases, $S_i / N k_B = \ln\left(\hat{c} V_i E_i^{3/2}\right), 1 \leq i \leq 2$. Here, $m$ denotes the mass of the wall, $2L$ is the total length of the system, and $V_i$ is the volume of the $i^{\text{th}}$ container. As in [16, 25] $N k_B = 1$ and $\alpha = 0.5$ fix the characteristic macroscopic unit of entropy while $\hat{c} = 102.25$ ensures the argument of the logarithm defining $E_i$ is dimensionless. This leads to the total entropy $S(\boldsymbol{x}) = S_1 + S_2$ and the total energy $E(\boldsymbol{x}) = (1/2m)p^2 + E_1(\boldsymbol{x}) + E_2(\boldsymbol{x})$, which are guaranteed to be nondecreasing and constant, respectively.

The primary goal here is to verify that NMS can accurately and stably predict gas container dynamics without the need to observe the entropic variables $S_1, S_2$. To that end, NMS has been compared to GNODE, SPNN, and GFINN on the task of predicting the trajectories of this metriplectic system over time, with results displayed in Table 2. More precisely, given an intial condition $\boldsymbol{x}_0$ and an interval $0 < t_{\text{train}} < t_{\text{valid}} < t_{\text{test}}$, each method is trained on partial state information (in the case of NMS) or full state information (in the case of the others) from the interval $[0, t_{\text{train}}]$ and validated on $(t_{\text{train}}, t_{\text{valid}}]$ before state errors in $q, p$ only are calculated on the whole interval $[0, t_{\text{test}}]$. As can be seen from Table 2 and Figure 2, NMS is remarkably accurate over unseen timescales even in this unfair comparison, avoiding the unphysical behavior which often hinders soft-constrained methods like SPNN. The energy and instantaneous entropy plots in Figure 2 further confirm that the strong enforcement of metriplectic structure guaranteed by NMS leads to correct energetic and entropic dynamics for all time.

## 5.2   Thermoelastic double pendulum

Next, consider the thermoelastic double pendulum from [26] with 10-dimensional state variable $\boldsymbol{x} = \begin{pmatrix} \boldsymbol{q}_1 & \boldsymbol{q}_2 & \boldsymbol{p}_1 & \boldsymbol{p}_2 & S_1 & S_2 \end{pmatrix}^\intercal$, which represents a highly challenging benchmark for metriplectic methods. The equations of motion in this case are given for $1 \leq i \leq 2$ as

$$\dot{\boldsymbol{q}}_i = \frac{\boldsymbol{p}_i}{m_i}, \quad \dot{\boldsymbol{p}}_i = -\partial_{\boldsymbol{q}_i}(E_1(\boldsymbol{x}) + E_2(\boldsymbol{x})), \quad \dot{S}_1 = \kappa\left(T_1^{-1} T_2 - 1\right), \quad \dot{S}_2 = \kappa\left(T_1 T_2^{-1} - 1\right),$$

where $\kappa > 0$ is a thermal conductivity constant (set to 1), $m_i$ is the mass of the $i^{\text{th}}$ spring (also set to 1) and $T_i = \partial_{S_i} E_i$ is its absolute temperature. In this case, $\boldsymbol{q}_i, \boldsymbol{p}_i \in \mathbb{R}^2$ represent the position and

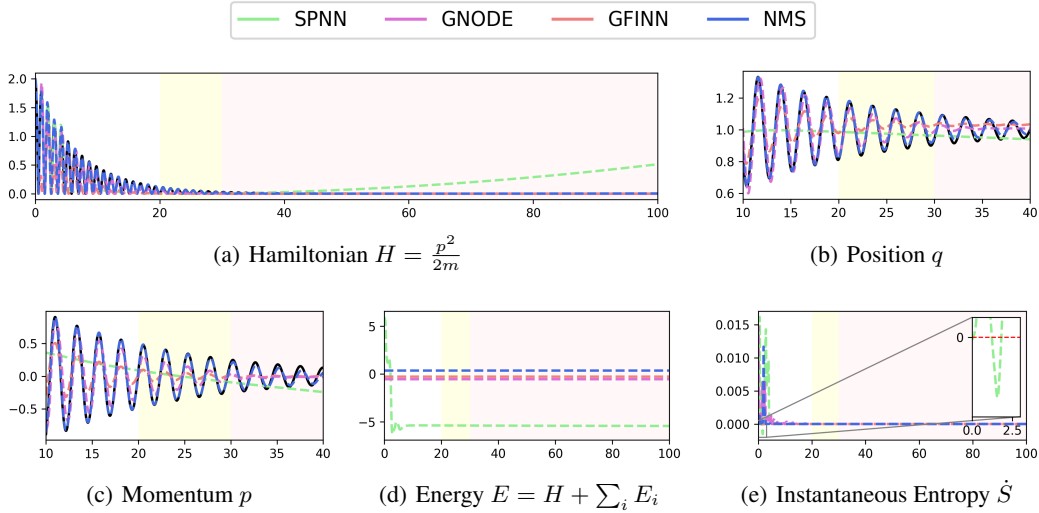

(a) Hamiltonian $H = \frac{p^2}{2m}$

(b) Position $q$

(c) Momentum $p$

(d) Energy $E = H + \sum_i E_i$

(e) Instantaneous Entropy $\dot{S}$

Figure 2: The ground-truth and predicted position, momentum, instantaneous entropy, and energies for the two gas containers example in the training (white), validation (yellow), and testing (red) regimes.

Table 2: Prediction errors for $\boldsymbol{x}^o$ measured in MSE and MAE on the interval $[0, t_{\text{test}}]$ in the two gas containers example (left) and on the test set in the thermoelastic double pendulum example (right).

|  | NODE | SPNN | GNODE | GFINN | NMS |  | NODE | SPNN | GNODE | GFINN | NMS |
|---|---|---|---|---|---|---|---|---|---|---|---|
| MSE | .12 ± .04 | .13 ± .10 | .16 ± .10 | .07 ± .03 | **.01 ± .02** | MSE | .41 ± .01 | .42 ± .01 | .42 ± .01 | .40 ± .03 | **.38 ± .03** |
| MAE | .25 ± .10 | .26 ± .14 | .25 ± .13 | .13 ± .03 | **.08 ± .06** | MAE | .48 ± .04 | .47 ± .03 | .46 ± .04 | .43 ± .07 | **.42 ± .07** |

momentum of the $i^{\text{th}}$ mass, while $S_i$ represents the entropy of the $i^{\text{th}}$ pendulum. As before, the total entropy $S(\boldsymbol{x}) = S_1 + S_2$ is the sum of the entropies of the two springs, while defining the internal energies $E_i(\boldsymbol{x}) = (1/2)(\ln \lambda_i)^2 + \ln \lambda_i + e^{S_i - \ln \lambda_i} - 1, \lambda_1 = |\boldsymbol{q}_i|, \lambda_2 = |\boldsymbol{q}_2 - \boldsymbol{q}_1|$, leads to the total energy $E(\boldsymbol{x}) = (1/2m_1)|\boldsymbol{p}_1|^2 + (1/2m_2)|\boldsymbol{p}_2|^2 + E_1(\boldsymbol{x}) + E_2(\boldsymbol{x})$.

The task in this case is prediction across initial conditions. As in [18], 100 trajectories are drawn from the ranges in Appendix B and integrated over the interval $[0, 40]$ with $\Delta t = 0.1$, with an 80/10/10 split for training/validation/testing. Here all compared models are trained using full state information. As seen in Table 2, NMS is again the most performant, although all models struggle to approximate the dynamics over the entire training interval. It is also notable that the training time of NMS is greatly decreased relative to GNODE and GFINN due to its improved quadratic scaling; a representative study to this effect is given in Appendix D.

# 6 Conclusion

Neural metriplectic systems (NMS) have been considered for learning finite-dimensional metriplectic dynamics from data. Making use of novel non-redundant parameterizations for metriplectic operators, NMS provably approximates arbitrary nondegenerate metriplectic systems with generalization error bounded in terms of the operator approximation quality. Benchmark examples have shown that NMS is both more scalable and more accurate than previous methods, including when only partial state information is observed. Future work will consider extensions of NMS to infinite-dimensional metriplectic systems with the aim of addressing its main limitation: the difficulty of scaling NMS (among all present methods for metriplectic learning) to realistic, 3-D problems of the size that would be considered in practice. A promising direction is to consider the use of NMS in model reduction, where sparse, large-scale systems are converted to small, dense systems through a clever choice of encoding/decoding.

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

## A  Proof of Theoretical Results

This Appendix provides proof of the analytical results in Section 3 of the body. First, the parameterizations of $L$, $M$ in terms of exterior algebra are established.

*Proof of Lemma 3.2.* First, it is necessary to check that the operators $L$, $M$ parameterized this way satisfy the symmetries and degeneracy conditions claimed in the statement. To that end, recall that $a \wedge b \simeq ab^\mathsf{T} - ba^\mathsf{T}$, meaning that $(ab^\mathsf{T} - ba^\mathsf{T})^\mathsf{T} \simeq b \wedge a = -a \wedge b$. It follows that $A^\mathsf{T} \simeq \tilde{A} = -A$ where $\tilde{A}$ denotes the reversion of A, i.e., $\tilde{A} = \sum_{i<j} A^{ij} e_j \wedge e_i$. Therefore, we may write

$$L^\mathsf{T} \simeq \tilde{A} - \frac{1}{|\nabla S|^2} \widetilde{A\nabla S \wedge \nabla S} = -A + \frac{1}{|\nabla S|^2} A\nabla S \wedge \nabla S \simeq -L,$$

showing that $L^\mathsf{T} = -L$. Moreover, using that

$$(b \wedge c) \cdot a = -a \cdot (b \wedge c) = (a \cdot c)b - (a \cdot b)c,$$

it follows that

$$L\nabla S = A \cdot \nabla S - \frac{1}{|\nabla S|^2}(A\nabla S \wedge \nabla S) \cdot \nabla S = A\nabla S - A\nabla S = 0,$$

since $\nabla S \cdot A\nabla S = -\nabla S \cdot A\nabla S = 0$. Moving to the case of $M$, notice that $M = D_{st} v^s \otimes v^t$ for a particular choice of $v$, meaning that

$$M^\mathsf{T} = \sum_{s,t} D_{st}(v^s \otimes v^t)^\mathsf{T} = \sum_{s,t} D_{st} v^t \otimes v^s = \sum_{t,s} D_{ts} v^s \otimes v^t = \sum_{s,t} D_{st} v^s \otimes v^t = M,$$

since $D$ is a symmetric matrix. Additionally, it is straightforward to check that, for any $1 \le s \le r$,

$$v^s \cdot \nabla E = \left( b^s - \frac{b^s \cdot \nabla E}{|\nabla E|^2} \nabla E \right) \cdot \nabla E = b^s \cdot \nabla E - b^s \cdot \nabla E = 0.$$

So, it follows immediately that

$$M\nabla E = \sum_{s,t} D_{st}(v^s \otimes v^t) \cdot \nabla E = \sum_{s,t} D_{st}(v^t \cdot \nabla E) v^s = 0.$$

Now, observe that

$$L = A - \frac{1}{|\nabla S|^2}(A\nabla S(\nabla S)^\mathsf{T} - \nabla S(A\nabla S)^\mathsf{T})$$

$$= A - \frac{1}{|\nabla S|^2}(A\nabla S(\nabla S^\mathsf{T}) + \nabla S(\nabla S)^\mathsf{T} A)$$

$$= \left( I - \frac{\nabla S(\nabla S)^\mathsf{T}}{|\nabla S|^2} \right) A \left( I - \frac{\nabla S(\nabla S)^\mathsf{T}}{|\nabla S|^2} \right) = P_S^\perp A P_S^\perp,$$

since $A^\mathsf{T} = -A$ and hence $v^\mathsf{T} A v = 0$ for all $v \in \mathbb{R}^n$. Similarly, it follows that for every $1 \le s \le r$,

$$P_E^\perp b^s = b^s - \frac{b^s \cdot \nabla E}{|\nabla E|^2} \nabla E,$$

and therefore $M$ is expressible as

$$M = \sum_{s,t} D_{st}(P_E^\perp b^s)(P_E^\perp b^t)^\mathsf{T} = P_E^\perp B D B^\mathsf{T} P_E^\perp. \quad \square$$

With Lemma 3.2 established, the proof of Theorem 3.4 is straightforward.

*Proof of Theorem 3.4.* The "if" direction follows immediately from Lemma 3.2. Now, suppose that $L$ and $M$ define a metriplectic system, meaning that the mentioned symmetries and degeneracy conditions hold. Then, it follows from $L\nabla S = 0$ that the projection $P_S^\perp L P_S^\perp = L$ leaves $L$ invariant, so that choosing $A = L$ yields $P_S^\perp A P_S^\perp = L$. Similarly, from positive semi-definiteness and $M\nabla E = 0$ it follows that $M = U\Lambda U^\mathsf{T} = P_E^\perp U \Lambda U^\mathsf{T} P_E^\perp$ for some column-orthonormal $U \in \mathbb{R}^{N \times r}$ and positive diagonal $\Lambda \in \mathbb{R}^{r \times r}$. Therefore, choosing $B = U$ and $D = \Lambda$ yields $M = P_E^\perp B D B^\mathsf{T} P^\perp$, as desired. $\square$

Looking toward the proof of Proposition 3.7, we also need to establish the following Lemmata which give control over the orthogonal projectors $\boldsymbol{P}^{\perp}_{\tilde{E}}, \boldsymbol{P}^{\perp}_{\tilde{S}}$. First, we recall how control over the $L^{\infty}$ norm $|\cdot|_{\infty}$ of a matrix field gives control over its spectral norm $|\cdot|$.

**Lemma A.1.** *Let $\boldsymbol{A} : K \to \mathbb{R}^{n \times n}$ be a matrix field defined on the compact set $K \subset \mathbb{R}^n$ with $m$ continuous derivatives. Then, for any $\varepsilon > 0$ there exists a two-layer neural network $\tilde{\boldsymbol{A}} : K \to \mathbb{R}^{n \times n}$ such that $\sup_{\boldsymbol{x} \in K} \left| \boldsymbol{A} - \tilde{\boldsymbol{A}} \right| < \varepsilon$ and $\sup_{\boldsymbol{x} \in K} \left| \nabla^k \boldsymbol{A} - \nabla^k \tilde{\boldsymbol{A}} \right|_{\infty} < \varepsilon$ for $1 \le k \le m$ where $\nabla^k$ is the (total) derivative operator of order $k$.*

*Proof.* This will be a direct consequence of Corollary 2.2 in [24] provided we show that $|\boldsymbol{A}| \le c |\boldsymbol{A}|_{\infty}$ for some $c > 0$. To that end, if $\sigma_1 \ge ... \ge \sigma_r > 0$ ($r \le n$) denote the nonzero singular values of $\boldsymbol{A} - \tilde{\boldsymbol{A}}$, it follows that for each $\boldsymbol{x} \in K$,

$$\left| \boldsymbol{A} - \tilde{\boldsymbol{A}} \right| = \sigma_1 \le \sqrt{\sigma_1^2 + ... + \sigma_r^2} = \sqrt{\sum_{i,j} \left| A_{ij} - \tilde{A}_{ij} \right|^2} = \left| \boldsymbol{A} - \tilde{\boldsymbol{A}} \right|_F.$$

On the other hand, it also follows that

$$\left| \boldsymbol{A} - \tilde{\boldsymbol{A}} \right|_F = \sqrt{\sum_{i,j} \left| A_{ij} - \tilde{A}_{ij} \right|^2} \le \sqrt{\sum_{i,j} \max_{i,j} \left| A_{ij} - \tilde{A}_{ij} \right|} = n \sqrt{\max_{i,j} \left| A_{ij} - \tilde{A}_{ij} \right|} = n \left| \boldsymbol{A} - \tilde{\boldsymbol{A}} \right|_{\infty},$$

and therefore the desired inequality holds with $c = n$. Now, for any $\varepsilon > 0$ it follows from [24] that there exists a two layer network $\tilde{\boldsymbol{A}}$ with $m$ continuous derivatives such that $\sup_{\boldsymbol{x} \in K} \left| \boldsymbol{A} - \tilde{\boldsymbol{A}} \right|_{\infty} < \varepsilon/n$ and $\sup_{\boldsymbol{x} \in K} \left| \nabla^k \boldsymbol{A} - \nabla^k \tilde{\boldsymbol{A}} \right|_{\infty} < \varepsilon/n < \varepsilon$ for all $1 \le k \le m$. Therefore, it follows that

$$\sup_{\boldsymbol{x} \in K} \left| \boldsymbol{A} - \tilde{\boldsymbol{A}} \right| \le n \sup_{\boldsymbol{x} \in K} \left| \boldsymbol{A} - \tilde{\boldsymbol{A}} \right|_{\infty} < n \frac{\varepsilon}{n} = \varepsilon,$$

completing the argument. $\square$

Next, we bound the deviation in the orthogonal projectors $\boldsymbol{P}^{\perp}_{\tilde{E}}, \boldsymbol{P}^{\perp}_{\tilde{S}}$.

**Lemma A.2.** *Let $f : \mathbb{R}^n \to \mathbb{R}$ be such that $\nabla f \ne \boldsymbol{0}$ on the compact set $K \subset \mathbb{R}^n$. For any $\varepsilon > 0$, there exists a two-layer neural network $\tilde{f} : K \to \mathbb{R}$ such that $\nabla \tilde{f} \ne \boldsymbol{0}$ on $K$, $\sup_{\boldsymbol{x} \in K} \left| f - \tilde{f} \right| < \varepsilon$, $\sup_{\boldsymbol{x} \in K} \left| \nabla f - \nabla \tilde{f} \right| < \varepsilon$, and $\sup_{\boldsymbol{x} \in K} \left| \boldsymbol{P}^{\perp}_{f} - \boldsymbol{P}^{\perp}_{\tilde{f}} \right| < \varepsilon$.*

*Proof.* Denote $\nabla f = \boldsymbol{v}$ and consider any $\tilde{\boldsymbol{v}} : K \to \mathbb{R}$. Since $|\boldsymbol{v}| \le |\tilde{\boldsymbol{v}}| + |\boldsymbol{v} - \tilde{\boldsymbol{v}}|$, it follows for all $\boldsymbol{x} \in K$ that whenever $|\boldsymbol{v} - \tilde{\boldsymbol{v}}| < (1/2) \inf_{\boldsymbol{x} \in K} |\boldsymbol{v}|$,

$$|\tilde{\boldsymbol{v}}| \ge |\boldsymbol{v}| - |\boldsymbol{v} - \tilde{\boldsymbol{v}}| > |\boldsymbol{v}| - \frac{1}{2} \inf_{\boldsymbol{x} \in K} |\boldsymbol{v}| > 0,$$

so that $\tilde{\boldsymbol{v}} \ne 0$ in $K$, and since the square function is monotonic,

$$\inf_{\boldsymbol{x} \in K} |\tilde{\boldsymbol{v}}|^2 \ge \inf_{\boldsymbol{x} \in K} \left( |\boldsymbol{v}| - \frac{1}{2} \inf_{\boldsymbol{x} \in K} |\boldsymbol{v}| \right)^2 = \frac{1}{4} \inf_{\boldsymbol{x} \in K} |\boldsymbol{v}|^2.$$

On the other hand, we also have $|\tilde{\boldsymbol{v}}| \le |\boldsymbol{v}| + |\tilde{\boldsymbol{v}} - \boldsymbol{v}| < |\boldsymbol{v}| + (1/2) \inf_{\boldsymbol{x} \in K} |\boldsymbol{v}|$, so that, adding and subtracting $\tilde{\boldsymbol{v}} \boldsymbol{v}^{\mathsf{T}}$ and applying Cauchy-Schwarz, it follows that for all $\boldsymbol{x} \in K$,

$$|\boldsymbol{v}\boldsymbol{v}^{\mathsf{T}} - \tilde{\boldsymbol{v}}\tilde{\boldsymbol{v}}^{\mathsf{T}}| \le |\boldsymbol{v} - \tilde{\boldsymbol{v}}||\boldsymbol{v}| + |\tilde{\boldsymbol{v}}||\boldsymbol{v} - \tilde{\boldsymbol{v}}| \le 2 \max\{|\boldsymbol{v}|, |\tilde{\boldsymbol{v}}|\} |\boldsymbol{v} - \tilde{\boldsymbol{v}}| < \left( 2|\boldsymbol{v}| + \inf_{\boldsymbol{x} \in K} |\boldsymbol{v}| \right) |\boldsymbol{v} - \tilde{\boldsymbol{v}}|.$$

Now, by Corollary 2.2 in [24], for any $\varepsilon > 0$ there exists a two-layer neural network $\tilde{f} : K \to \mathbb{R}$ such that

$$\sup_{\boldsymbol{x} \in K} \left| \boldsymbol{v} - \nabla \tilde{f} \right| < \min \left\{ \frac{1}{2} \inf_{\boldsymbol{x} \in K} |\boldsymbol{v}|, \frac{\inf_{\boldsymbol{x} \in K} |\boldsymbol{v}|^2}{2 \sup_{\boldsymbol{x} \in K} |\boldsymbol{v}| + \inf_{\boldsymbol{x} \in K} |\boldsymbol{v}|} \frac{\varepsilon}{4}, \varepsilon \right\} \le \varepsilon,$$

and also $\sup_{\boldsymbol{x}\in K}\left|f-\tilde{f}\right|<\varepsilon$. Letting $\tilde{\boldsymbol{v}}=\nabla\tilde{f}$, it follows that for all $\boldsymbol{x}\in K$,

$$\left|\boldsymbol{P}_f^\perp-\boldsymbol{P}_{\tilde{f}}^\perp\right|=\left|\frac{\boldsymbol{v}\boldsymbol{v}^\mathsf{T}}{|\boldsymbol{v}|^2}-\frac{\tilde{\boldsymbol{v}}\tilde{\boldsymbol{v}}^\mathsf{T}}{|\tilde{\boldsymbol{v}}|^2}\right|\leq\frac{|\boldsymbol{v}\boldsymbol{v}^\mathsf{T}-\tilde{\boldsymbol{v}}\tilde{\boldsymbol{v}}^\mathsf{T}|}{\min\left\{|\boldsymbol{v}|^2,|\tilde{\boldsymbol{v}}|^2\right\}}\leq\frac{2|\boldsymbol{v}|+\inf_{\boldsymbol{x}\in K}|\boldsymbol{v}|}{\min\left\{|\boldsymbol{v}|^2,|\tilde{\boldsymbol{v}}|^2\right\}}|\boldsymbol{v}-\tilde{\boldsymbol{v}}|,$$

and therefore, taking the supremum of both sides and applying the previous work yields the desired estimate,

$$\sup_{\boldsymbol{x}\in K}\left|\boldsymbol{P}_f^\perp-\boldsymbol{P}_{\tilde{f}}^\perp\right|\leq4\frac{2\sup_{\boldsymbol{x}\in K}|\boldsymbol{v}|+\inf_{\boldsymbol{x}\in K}|\boldsymbol{v}|}{\inf_{\boldsymbol{x}\in K}|\boldsymbol{v}|^2}\sup_{\boldsymbol{x}\in K}|\boldsymbol{v}-\tilde{\boldsymbol{v}}|<\varepsilon.\qquad\square$$

With these intermediate results established, the proof of the approximation result Proposition 3.7 proceeds as follows.

*Proof of Proposition 3.7.* Recall from Theorem 3.4 that we can write $\boldsymbol{L}=\boldsymbol{P}_S^\perp(\boldsymbol{A}_{\text{tri}}-\boldsymbol{A}_{\text{tri}}^\mathsf{T})\boldsymbol{P}_S^\perp$ and similarly for $\tilde{\boldsymbol{L}}$. Notice that, by adding and subtracting $\boldsymbol{P}_{\tilde{S}}^\perp\boldsymbol{A}_{\text{tri}}\boldsymbol{P}_S^\perp$ and $\boldsymbol{P}_{\tilde{S}}^\perp\tilde{\boldsymbol{A}}_{\text{tri}}\boldsymbol{P}_S^\perp$, it follows that for all $\boldsymbol{x}\in K$,

$$\left|\boldsymbol{P}_S^\perp\boldsymbol{A}_{\text{tri}}\boldsymbol{P}_S^\perp-\boldsymbol{P}_{\tilde{S}}^\perp\tilde{\boldsymbol{A}}_{\text{tri}}\boldsymbol{P}_{\tilde{S}}^\perp\right|$$

$$=\left|(\boldsymbol{P}_S^\perp-\boldsymbol{P}_{\tilde{S}}^\perp)\boldsymbol{A}_{\text{tri}}\boldsymbol{P}_S^\perp+\boldsymbol{P}_{\tilde{S}}^\perp\left(\boldsymbol{A}_{\text{tri}}-\tilde{\boldsymbol{A}}_{\text{tri}}\right)\boldsymbol{P}_S^\perp+\boldsymbol{P}_{\tilde{S}}^\perp\tilde{\boldsymbol{A}}_{\text{tri}}(\boldsymbol{P}_S^\perp-\boldsymbol{P}_{\tilde{S}}^\perp)\right|$$

$$\leq\left|\boldsymbol{P}_S^\perp-\boldsymbol{P}_{\tilde{S}}^\perp\right||\boldsymbol{A}_{\text{tri}}|+\left|\boldsymbol{A}_{\text{tri}}-\tilde{\boldsymbol{A}}_{\text{tri}}\right|+\left|\tilde{\boldsymbol{A}}_{\text{tri}}\right|\left|\boldsymbol{P}_S^\perp-\boldsymbol{P}_{\tilde{S}}^\perp\right|$$

$$\leq2\max\left\{|\boldsymbol{A}_{\text{tri}}|,\left|\tilde{\boldsymbol{A}}_{\text{tri}}\right|\right\}\left|\boldsymbol{P}_S^\perp-\boldsymbol{P}_{\tilde{S}}^\perp\right|+\left|\boldsymbol{A}_{\text{tri}}-\tilde{\boldsymbol{A}}_{\text{tri}}\right|$$

where we have used that $\boldsymbol{P}_S^\perp,\boldsymbol{P}_{\tilde{S}}^\perp$ have unit spectral norm. By Lemma A.1, for any $\varepsilon>0$ there exists a two layer neural network $\tilde{\boldsymbol{A}}_{\text{tri}}$ such that $\sup_{\boldsymbol{x}\in K}\left|\boldsymbol{A}_{\text{tri}}-\tilde{\boldsymbol{A}}_{\text{tri}}\right|<\frac{\varepsilon}{4}$, and by Lemma A.2 there exists a two-layer network $\tilde{S}$ with $\nabla\tilde{S}\neq\boldsymbol{0}$ on $K$ such that

$$\sup_{\boldsymbol{x}\in K}\left|\boldsymbol{P}_S^\perp-\boldsymbol{P}_{\tilde{S}}^\perp\right|<\min\left\{\varepsilon,\max\left\{\sup_{\boldsymbol{x}\in K}|\boldsymbol{A}_{\text{tri}}|,\sup_{\boldsymbol{x}\in K}\left|\tilde{\boldsymbol{A}}_{\text{tri}}\right|\right\}^{-1}\frac{\varepsilon}{8}\right\}.$$

It follows that $\tilde{S},\nabla\tilde{S}$ are $\varepsilon$-close to $S,\nabla S$ on $K$ and

$$\sup_{\boldsymbol{x}\in K}\left(2\max\left\{|\boldsymbol{A}_{\text{tri}}|,\left|\tilde{\boldsymbol{A}}_{\text{tri}}\right|\right\}\left|\boldsymbol{P}_S^\perp-\boldsymbol{P}_{\tilde{S}}^\perp\right|\right)<\frac{\varepsilon}{4}.$$

Therefore, the estimate

$$\sup_{\boldsymbol{x}\in K}\left|\boldsymbol{L}-\tilde{\boldsymbol{L}}\right|\leq2\sup_{\boldsymbol{x}\in K}\left|\boldsymbol{P}_S^\perp\boldsymbol{A}_{\text{tri}}\boldsymbol{P}_S^\perp-\boldsymbol{P}_{\tilde{S}}^\perp\tilde{\boldsymbol{A}}_{\text{tri}}\boldsymbol{P}_{\tilde{S}}^\perp\right|<2\left(\frac{\varepsilon}{4}+\frac{\varepsilon}{4}\right)=\varepsilon,$$

implies that $\tilde{\boldsymbol{L}}$ is $\varepsilon$-close to $\boldsymbol{L}$ on $K$ as well.

Moving to the case of $\boldsymbol{M}$, we see that for all $\boldsymbol{x}\in K$, by writing $\boldsymbol{M}=\boldsymbol{U}\boldsymbol{\Lambda}\boldsymbol{U}^\mathsf{T}=\boldsymbol{K}_{\text{chol}}\boldsymbol{K}_{\text{chol}}^\mathsf{T}$ for $\boldsymbol{K}_{\text{chol}}=\boldsymbol{U}\boldsymbol{\Lambda}^{1/2}$ and repeating the first calculation with $\boldsymbol{K}_{\text{chol}}$ in place of $\boldsymbol{A}_{\text{tri}}$ and $\boldsymbol{P}_E^\perp$ in place of $\boldsymbol{P}_S^\perp$,

$$\left|\boldsymbol{P}_E^\perp\boldsymbol{K}_{\text{chol}}\boldsymbol{K}_{\text{chol}}^\mathsf{T}\boldsymbol{P}_E^\perp-\boldsymbol{P}_{\tilde{E}}^\perp\tilde{\boldsymbol{K}}_{\text{chol}}\tilde{\boldsymbol{K}}_{\text{chol}}^\mathsf{T}\boldsymbol{P}_{\tilde{E}}^\perp\right|$$

$$\leq2\max\left\{|\boldsymbol{K}_{\text{chol}}|,\left|\tilde{\boldsymbol{K}}_{\text{chol}}\right|\right\}\left|\boldsymbol{P}_E^\perp-\boldsymbol{P}_{\tilde{E}}^\perp\right|+\left|\boldsymbol{K}_{\text{chol}}\boldsymbol{K}_{\text{chol}}^\mathsf{T}-\tilde{\boldsymbol{K}}_{\text{chol}}\tilde{\boldsymbol{K}}_{\text{chol}}^\mathsf{T}\right|.$$

Moreover, if $\left|\boldsymbol{K}_{\text{chol}}-\tilde{\boldsymbol{K}}_{\text{chol}}\right|<(1/2)\inf_{\boldsymbol{x}\in K}|\boldsymbol{K}_{\text{chol}}|$ for all $\boldsymbol{x}\in K$ then similar arguments as used in the proof of Lemma A.2 yield the following estimate for all $\boldsymbol{x}\in K$,

$$\left|\boldsymbol{K}_{\text{chol}}\boldsymbol{K}_{\text{chol}}^\mathsf{T}-\tilde{\boldsymbol{K}}_{\text{chol}}\tilde{\boldsymbol{K}}_{\text{chol}}^\mathsf{T}\right|\leq2\max\left\{|\boldsymbol{K}_{\text{chol}}|,\left|\tilde{\boldsymbol{K}}_{\text{chol}}\right|\right\}\left|\boldsymbol{K}_{\text{chol}}-\tilde{\boldsymbol{K}}_{\text{chol}}\right|$$

$$\leq\left(2|\boldsymbol{K}_{\text{chol}}|+\inf_{\boldsymbol{x}\in K}|\boldsymbol{K}_{\text{chol}}|\right)\left|\boldsymbol{K}_{\text{chol}}-\tilde{\boldsymbol{K}}_{\text{chol}}\right|.$$

As before, we now invoke Lemma A.1 to construct a two-layer lower-triangular network $\tilde{K}_{\mathrm{chol}}$ such that

$$\sup_{\boldsymbol{x}\in K}\left|\boldsymbol{K}_{\mathrm{chol}} - \tilde{\boldsymbol{K}}_{\mathrm{chol}}\right| < \min\left\{\frac{1}{2}\inf_{\boldsymbol{x}\in K}|\boldsymbol{K}_{\mathrm{chol}}|, \left(2\sup_{\boldsymbol{x}\in K}|\boldsymbol{K}_{\mathrm{chol}}| + \inf_{\boldsymbol{x}\in K}|\boldsymbol{K}_{\mathrm{chol}}|\right)^{-1}\frac{\varepsilon}{2}\right\},$$

as well as (using Lemma A.2) a network $\tilde{E}$ satisfying $\nabla\tilde{E}\neq\boldsymbol{0}$ on $K$ and

$$\sup_{\boldsymbol{x}\in K}\left|\boldsymbol{P}_E^\perp - \boldsymbol{P}_{\tilde{E}}^\perp\right| < \min\left\{\varepsilon, \max\left\{\sup_{\boldsymbol{x}\in K}|\boldsymbol{K}_{\mathrm{chol}}|, \sup_{\boldsymbol{x}\in K}\left|\tilde{\boldsymbol{K}}_{\mathrm{chol}}\right|\right\}^{-1}\frac{\varepsilon}{4}\right\}.$$

Again, it follows that $\tilde{E}, \nabla\tilde{E}$ are $\varepsilon$-close to $E, \nabla E$ on $K$, and by the work above we conclude

$$\sup_{\boldsymbol{x}\in K}\left|\boldsymbol{M} - \tilde{\boldsymbol{M}}\right| = \sup_{\boldsymbol{x}\in K}\left|\boldsymbol{P}_E^\perp\boldsymbol{K}_{\mathrm{chol}}\boldsymbol{K}_{\mathrm{chol}}^\intercal\boldsymbol{P}_E^\perp - \boldsymbol{P}_{\tilde{E}}^\perp\tilde{\boldsymbol{K}}_{\mathrm{chol}}\tilde{\boldsymbol{K}}_{\mathrm{chol}}^\intercal\boldsymbol{P}_{\tilde{E}}^\perp\right| < \frac{\varepsilon}{2} + \frac{\varepsilon}{2} = \varepsilon,$$

as desired. $\qquad\square$

It is now possible to give a proof of the error bound in Theorem 3.9. Recall the $L^2([0,T])$ error metric $\|\boldsymbol{x}\|$ and Lipschitz constant $L_f$, defined for all $\boldsymbol{x}, \boldsymbol{y} \in \mathbb{R}^n$ and Lipschitz continuous functions $f$ as

$$\|\boldsymbol{x}\|^2 = \int_0^T |\boldsymbol{x}|^2\, dt, \quad |f(\boldsymbol{x}) - f(\boldsymbol{y})| \le L_f|\boldsymbol{x} - \boldsymbol{y}|.$$

*Proof of Theorem 3.9.* First, note that the assumption that one of $E, -S$ (without loss of generality, say $E$) has bounded sublevel sets implies bounded trajectories for the state $\boldsymbol{x}$ as in Remark 3.8, so we may assume $\boldsymbol{x} \in K$ for some compact $K \subset \mathbb{R}^n$. Moreover, for any $\varepsilon > 0$ it follows from Proposition 3.7 that there are approximate networks $\tilde{E}, \tilde{S}$ which are $\varepsilon$-close to $E, S$ on $K$. Additionally, it follows that $\tilde{E}, \tilde{S}$ have nonzero gradients $\nabla\tilde{E}, \nabla\tilde{S}$ which are also $\varepsilon$-close to the true gradients $\nabla E, \nabla S$ on $K$. This implies that for each $\boldsymbol{x} \in K$, $E = \tilde{E} + (E - \tilde{E}) \le \tilde{E} + \varepsilon$, so it follows that the sublevel sets $\{\boldsymbol{x} \mid \tilde{E}(\boldsymbol{x}) \le m\} \subseteq \{\boldsymbol{x} \mid E(\boldsymbol{x}) \le m + \varepsilon\}$ are also bounded. Therefore, we may assume (by potentially enlarging $K$) that both $\boldsymbol{x}, \tilde{\boldsymbol{x}} \in K$ lie in the compact set $K$ for all time.

Now, let $\boldsymbol{y} = \boldsymbol{x} - \tilde{\boldsymbol{x}}$. The next goal is to bound the following quantity:

$$|\dot{\boldsymbol{y}}| = \left|\boldsymbol{L}(\boldsymbol{x})\nabla E(\boldsymbol{x}) + \boldsymbol{M}(\boldsymbol{x})\nabla S(\boldsymbol{x}) - \tilde{\boldsymbol{L}}(\tilde{\boldsymbol{x}})\nabla\tilde{E}(\tilde{\boldsymbol{x}}) - \tilde{\boldsymbol{M}}(\tilde{\boldsymbol{x}})\nabla\tilde{S}(\tilde{\boldsymbol{x}})\right|$$

$$= \left|\left(\boldsymbol{L}(\boldsymbol{x})\nabla E(\boldsymbol{x}) - \tilde{\boldsymbol{L}}(\tilde{\boldsymbol{x}})\nabla E(\tilde{\boldsymbol{x}})\right) + \left(\boldsymbol{M}(\boldsymbol{x})\nabla S(\boldsymbol{x}) - \tilde{\boldsymbol{M}}(\tilde{\boldsymbol{x}})\nabla S(\tilde{\boldsymbol{x}})\right)\right| =: |\dot{\boldsymbol{y}}_E + \dot{\boldsymbol{y}}_S|.$$

To that end, notice that by adding and subtracting $\boldsymbol{L}(\boldsymbol{x})\nabla E(\tilde{\boldsymbol{x}}), \tilde{\boldsymbol{L}}(\boldsymbol{x})\nabla E(\tilde{\boldsymbol{x}}), \tilde{\boldsymbol{L}}(\tilde{\boldsymbol{x}})\nabla E(\tilde{\boldsymbol{x}})$, it follows that

$$\dot{\boldsymbol{y}}_E = \boldsymbol{L}(\boldsymbol{x})(\nabla E(\boldsymbol{x}) - \nabla E(\tilde{\boldsymbol{x}})) + \left(\boldsymbol{L}(\boldsymbol{x}) - \tilde{\boldsymbol{L}}(\boldsymbol{x})\right)\nabla E(\tilde{\boldsymbol{x}})$$

$$+ \left(\tilde{\boldsymbol{L}}(\boldsymbol{x}) - \tilde{\boldsymbol{L}}(\tilde{\boldsymbol{x}})\right)\nabla E(\tilde{\boldsymbol{x}}) + \tilde{\boldsymbol{L}}(\tilde{\boldsymbol{x}})\left(\nabla E(\tilde{\boldsymbol{x}}) - \nabla\tilde{E}(\tilde{\boldsymbol{x}})\right).$$

By Proposition 3.7 there exists a two-layer neural network $\tilde{\boldsymbol{L}}$ with one continuous derivative such that $\sup_{\boldsymbol{x}\in K}\left|\boldsymbol{L} - \tilde{\boldsymbol{L}}\right| < \varepsilon$, which implies that $\tilde{\boldsymbol{L}}$ is Lipschitz continuous with (uniformly well-approximated) Lipschitz constant. Using this fact along with the assumed Lipschitz continuity of $\nabla E$ and the approximation properties of the network $\tilde{E}$ already constructed then yields

$$|\dot{\boldsymbol{y}}_E| \le \left(L_{\nabla E}\sup_{\boldsymbol{x}\in K}|\boldsymbol{L}| + L_{\tilde{\boldsymbol{L}}}\sup_{\boldsymbol{x}\in K}|\nabla E|\right)|\boldsymbol{y}| + \varepsilon\left(\sup_{\boldsymbol{x}\in K}\left|\tilde{\boldsymbol{L}}\right| + \sup_{\boldsymbol{x}\in K}|\nabla E|\right) =: a_E|\boldsymbol{y}| + \varepsilon\, b_E.$$

Similarly, by adding and subtracting $\boldsymbol{M}(\boldsymbol{x})\nabla S(\tilde{\boldsymbol{x}}), \tilde{\boldsymbol{M}}(\boldsymbol{x})\nabla S(\tilde{\boldsymbol{x}}), \tilde{\boldsymbol{M}}(\tilde{\boldsymbol{x}})\nabla S(\tilde{\boldsymbol{x}})$, it follows that

$$\dot{\boldsymbol{y}}_S = \boldsymbol{M}(\boldsymbol{x})(\nabla S(\boldsymbol{x}) - \nabla S(\tilde{\boldsymbol{x}})) + \left(\boldsymbol{M}(\boldsymbol{x}) - \tilde{\boldsymbol{M}}(\boldsymbol{x})\right)\nabla S(\tilde{\boldsymbol{x}})$$

$$+ \left(\tilde{\boldsymbol{M}}(\boldsymbol{x}) - \tilde{\boldsymbol{M}}(\tilde{\boldsymbol{x}})\right)\nabla S(\tilde{\boldsymbol{x}}) + \tilde{\boldsymbol{M}}(\tilde{\boldsymbol{x}})\left(\nabla S(\tilde{\boldsymbol{x}}) - \nabla\tilde{S}(\tilde{\boldsymbol{x}})\right).$$

By Proposition 3.7, there exists a two-layer network $\tilde{M}$ with one continuous derivative such that $\sup_{\boldsymbol{x} \in K} \left| \boldsymbol{M} - \tilde{\boldsymbol{M}} \right| < \varepsilon$, with $\tilde{\boldsymbol{M}}$ Lipschitz continuous for the same reason as before. It follows from this and $\sup_{\boldsymbol{x} \in K} \left| \nabla S - \nabla \tilde{S} \right| < \varepsilon$ that

$$|\dot{\boldsymbol{y}}_S| \leq \left( L_{\nabla S} \sup_{\boldsymbol{x} \in K} |\boldsymbol{M}| + L_{\tilde{\boldsymbol{M}}} \sup_{\boldsymbol{x} \in K} |\nabla S| \right) |\boldsymbol{y}| + \varepsilon \left( \sup_{\boldsymbol{x} \in K} \left| \tilde{\boldsymbol{M}} \right| + \sup_{\boldsymbol{x} \in K} |\nabla S| \right) =: a_S |\boldsymbol{y}| + \varepsilon \, b_S.$$

Now, recall that $\partial_t |\boldsymbol{y}| = |\boldsymbol{y}|^{-1} (\dot{\boldsymbol{y}} \cdot \boldsymbol{y}) \leq |\dot{\boldsymbol{y}}|$ by Cauchy-Schwarz, and therefore the time derivative of $|\boldsymbol{y}|$ is bounded by

$$\partial_t |\boldsymbol{y}| \leq |\dot{\boldsymbol{y}}_E| + |\dot{\boldsymbol{y}}_S| = (a_E + a_S) |\boldsymbol{y}| + \varepsilon (b_E + b_S) =: a |\boldsymbol{y}| + b.$$

This implies that $\partial_t |\boldsymbol{y}| - a |\boldsymbol{y}| \leq b$, so multiplying by the integrating factor $e^{-at}$ and integrating in time yields

$$|\boldsymbol{y}(t)| \leq \varepsilon b \int_0^t e^{a(t-\tau)} \, d\tau = \varepsilon \frac{b}{a} \left( e^{at} - 1 \right),$$

where we used that $\boldsymbol{y}(0) = \boldsymbol{0}$ since the initial condition of the trajectories is shared. Therefore, the $L^2$ error in time can be approximated by

$$\|\boldsymbol{y}\|^2 = \int_0^T |\boldsymbol{y}|^2 \, dt \leq \varepsilon^2 \frac{b^2}{a^2} \left( e^{2aT} - 2e^{aT} + T + 1 \right),$$

establishing the conclusion. $\qquad\square$

## B  Experimental and Implementation Details

This Appendix records additional details related to the numerical experiments in Section 5. For each benchmark problem, a set of trajectories is manufactured given initial conditions by simulating ODEs with known metriplectic structure. For the experiments in Table 2, only the observable variables are used to construct datasets, since entropic information is assumed to be unknown. Algorithm 2 summarizes the training of the dynamics models used for comparison with NMS.

---

**Algorithm 2** Training dynamics models

---

1: **Input:** snapshot data $\boldsymbol{X} \in \mathbb{R}^{n \times n_s}$, each column $\boldsymbol{x}_s = \boldsymbol{x}(t_s, \boldsymbol{\mu}_s)$, target rank $r \geq 1$
2: Initialize loss $L = 0$ and networks with parameters $\Theta$
3: **for** step in $N_{\text{steps}}$ **do**
4:     Randomly draw an initial condition $(t_{0_k}, \boldsymbol{x}_{0_k})$ where $k \in n_s$
5:     $\tilde{\boldsymbol{x}}_1, ..., \tilde{\boldsymbol{x}}_l = \text{ODEsolve}(\boldsymbol{x}_{0_k}, \dot{\boldsymbol{x}}, t_1, ..., t_l)$
6:     Compute the loss $L((\boldsymbol{x}_1^{\text{o}}, \ldots, \boldsymbol{x}_l^{\text{o}}), (\tilde{\boldsymbol{x}}_0^{\text{o}}, \ldots, \tilde{\boldsymbol{x}}_l^{\text{o}}))$
7:     Update the model parameters $\Theta$ via SGD
8: **end for**

---

For each compared method, integrating the ODEs is done via the Dormand–Prince method (do-pri5) [27] with relative tolerance $10^{-7}$ and absolute tolerance $10^{-9}$. The loss is evaluated by measuring the discrepancy between the ground truth observable states $\boldsymbol{x}^{\text{o}}$ and the approximate observable states $\tilde{\boldsymbol{x}}^{\text{o}}$ in the mean absolute error (MAE) metric. The model parameters $\Theta$ (i.e., the weights and biases) are updated by using Adamax [28] with an initial learning rate of 0.01. The number of training steps is set as 30,000, and the model parameters resulting in the best performance for the validation set are chosen for testing. Specific information related to the experiments in Section 5 is given in the subsections below.

For generating the results reported in Table 2, we implemented the proposed algorithm in Python 3.9.12 and PyTorch 2.0.0. Other required information is provided with the accompanying code. All experiments are conducted on Apple M2 Max chips with 96 GB memory. To provide the mean and the standard deviation, experiments are repeated three times with varying random seeds for all considered methods.

## B.1 Two gas containers

As mentioned in the body, the two gas container (TGC) problem tests models' predictive capability (i.e., extrapolation in time). To this end, one simulated trajectory is obtained by solving an IVP with a known TGC system and an initial condition, and the trajectory of the observable variables is split into three subsequences, $[0, t_{\text{train}}]$, $(t_{\text{train}}, t_{\text{val}}]$, and $(t_{\text{val}}, t_{\text{test}}]$ for training, validation, and test with $0 < t_{\text{train}} < t_{\text{val}} < t_{\text{test}}$.

In the experiment, a sequence of 100,000 timesteps is generated using the Runge–Kutta 4th-order (RK4) time integrator with a step size 0.001. The initial condition is given as $\boldsymbol{x} = (1, 2, 103.2874, 103.2874)$ following [29]. The training/validation/test split is defined by $t_{\text{train}} = 20$, $t_{\text{val}} = 30$, and $t_{\text{test}} = 100$. For a fair comparison, all considered models are set to have a similar number of model parameters, $\sim 2,000$. The specifications of the network architectures are:

- NMS: The total number of model parameters is 1959. The functions $\boldsymbol{A}_{\text{tri}}, \boldsymbol{B}, \boldsymbol{K}_{\text{chol}}, E, S$ are parameterized as MLPs with the Tanh nonlinear activation function. The MLPs parameterizing $\boldsymbol{A}_{\text{tri}}, \boldsymbol{B}, \boldsymbol{K}_{\text{chol}}, E$ are specified as 1 hidden layer with 10 neurons, and the on parameterizing $S$ is specified as 3 hidden layers with 25 neurons.

- NODE: The total number of model parameters is 2179. The black-box NODE is parameterized as an MLP with the Tanh nonlinear activation function, 4 hidden layers and 25 neurons.

- SPNN: The total number of model parameters is 1954. The functions $E$ and $S$ are parameterized as MLPs with the Tanh nonlinear activation function; each MLP is specified as 3 hidden layers and 20 neurons. The two 2-tensors defining $\boldsymbol{L}$ and $\boldsymbol{M}$ are defined as learnable $3 \times 3$ matrices.

- GNODE: The total number of model parameters is 2343. The functions $E$ and $S$ are parameterized as MLPs with the Tanh nonlinear activaton function; each MLP is specified as 2 hidden layers and 30 neurons. The matrices and 3-tensors required to learn $\boldsymbol{L}$ and $\boldsymbol{M}$ are defined as learnable $3 \times 3$ matrices and $3 \times 3 \times 3$ tensor.

- GFINN: The total number of model parameters is 2065. The functions $E$ and $S$ are parameterized as MLPs with Tanh nonlinear activation function; each MLP is specified as 2 hidden layers and 20 neurons. The matrices to required to learn $\boldsymbol{L}$ and $\boldsymbol{M}$ are defined as $K$ learnable $3 \times 3$ matrices, where $K$ is set to 2.

## B.2 Thermoelastic double pendulum

The equations of motion in this case are given for $1 \leq i \leq 2$ as

$$\dot{\boldsymbol{q}}_i = \frac{\boldsymbol{p}_i}{m_i}, \quad \dot{\boldsymbol{p}}_i = -\partial_{\boldsymbol{q}_i}(E_1(\boldsymbol{x}) + E_2(\boldsymbol{x})), \quad \dot{S}_1 = \kappa\big(T_1^{-1}T_2 - 1\big), \quad \dot{S}_2 = \kappa\big(T_1 T_2^{-1} - 1\big),$$

where $\kappa > 0$ is a thermal conductivity constant (set to 1), $m_i$ is the mass of the $i^{\text{th}}$ spring (also set to 1) and $T_i = \partial_{S_i} E_i$ is its absolute temperature. In this case, $\boldsymbol{q}_i, \boldsymbol{p}_i \in \mathbb{R}^2$ represent the position and momentum of the $i^{\text{th}}$ mass, while $S_i$ represents the entropy of the $i^{\text{th}}$ pendulum. As before, the total entropy $S(\boldsymbol{x}) = S_1 + S_2$ is the sum of the entropies of the two springs, while defining the internal energies

$$E_i(\boldsymbol{x}) = \frac{1}{2}(\ln \lambda_i)^2 + \ln \lambda_i + e^{S_i - \ln \lambda_i} - 1, \quad \lambda_1 = |\boldsymbol{q}_i|, \quad \lambda_2 = |\boldsymbol{q}_2 - \boldsymbol{q}_1|,$$

leads to the total energy $E(\boldsymbol{x}) = (1/2m_1)|\boldsymbol{p}_1|^2 + (1/2m_2)|\boldsymbol{p}_2|^2 + E_1(\boldsymbol{x}) + E_2(\boldsymbol{x})$.

The thermoelastic double pendulum experiment tests model prediction across initial conditions. In this case, 100 trajectories are generated by varying initial conditions that are randomly sampled from $[0.1,1.1] \times [-0.1,0.1] \times [2.1, 2.3] \times [-0.1,0.1] \times [-1.9,2.1] \times [0.9,1.1] \times [-0.1, 0.1] \times [0.9,1.1] \times [0.1,0.3] \subset \mathbb{R}^{10}$. Each trajectory is obtained from the numerical integration of the ODEs using an RK4 time integrator with step size 0.02 and the final time $T = 40$, resulting in the trajectories of length 2,000. The resulting 100 trajectories are split into 80/10/10 for training/validation/test sets. For a fair comparison, all considered models are again set to have similar number of model parameters, $\sim 2,000$. The specifications of the network architectures are:

- **NMS:** The total number of model parameters is 2201. The functions $\boldsymbol{A}, \boldsymbol{B}, \boldsymbol{K}, E, S$ are parameterized as MLPs with the Tanh nonlinear activation function. The MLPs parameterizing are specified as 1 hidden layer with 15 neurons.

- **NODE:** The total number of model parameters is 2005. The black-box NODE is parameterized as an MLP with the Tanh nonlinear activation function, 2 hidden layers and 35 neurons.

- **SPNN:** The total number of model parameters is 2362. The functions $E$ and $S$ are parameterized as MLPs with the Tanh nonlinear activation function; each MLP is specified as 3 hidden layers and 20 neurons. The two 2-tensors defining $\boldsymbol{L}$ and $\boldsymbol{M}$ are defined as learnable $3 \times 3$ matrices.

- **GNODE:** The total number of model parameters is 2151. The functions $E$ and $S$ are parameterized as MLPs with the Tanh nonlinear activaton function; each MLP is specified as 2 hidden layers and 15 neurons. The matrices and 3-tensors required to learn $\boldsymbol{L}$ and $\boldsymbol{M}$ are defined as learnable $3 \times 3$ matrices and $3 \times 3 \times 3$ tensor.

- **GFINN:** The total number of model parameters is 2180. The functions $E$ and $S$ are parameterized as MLPs with Tanh nonlinear activation function; each MLP is specified as 2 hidden layers and 15 neurons. The matrices to required to learn $\boldsymbol{L}$ and $\boldsymbol{M}$ are defined as $K$ learnable $3 \times 3$ matrices, where $K$ is set to 2.

## C Additional experiment: Damped nonlinear oscillator

Consider a damped nonlinear oscillator of variable dimension with state $\boldsymbol{x} = (\boldsymbol{q} \quad \boldsymbol{p} \quad S)^{\mathsf{T}}$, whose motion is governed by the metriplectic system

$$\dot{\boldsymbol{q}} = \frac{\boldsymbol{p}}{m}, \quad \dot{\boldsymbol{p}} = k \sin \boldsymbol{q} - \gamma \boldsymbol{p}, \quad \dot{S} = \frac{\gamma |\boldsymbol{q}|^2}{mT}.$$

Here $\boldsymbol{q}, \boldsymbol{p} \in \mathbb{R}^n$ denote the position and momentum of the oscillator, $S$ is the entropy of a surrounding thermal bath, and the constant parameters $m, \gamma, T$ are the mass, damping rate, and (constant) temperature. This leads to the total energy $E(\boldsymbol{x}) = (1/2m)|\boldsymbol{p}|^2 - k \cos \boldsymbol{q} + TS$, which is readily seen to be constant along solutions $\boldsymbol{x}(t)$.

It is now verified that NMS can accurately and stably predict the dynamics of a nonlinear oscillator $\boldsymbol{x} = (\boldsymbol{q} \quad \boldsymbol{p} \quad S)^{\mathsf{T}}$ in the case that $n = 1, 2$, both when the entropy $S$ is observable as well as when it is not. As before, the task considered is prediction in time, although all compared methods NODE, GNODE, and $\text{NMS}_{\text{known}}$ are now trained on full state information from the training interval, and test errors are computed over the full state $\boldsymbol{x}$ on the extrapolation interval $(t_{\text{valid}}, t_{\text{test}}]$, which is 150% longer than the training interval. In addition, another NMS model, $\text{NMS}_{\text{diff}}$, was trained using only the partial state information $\boldsymbol{x}^o = (\boldsymbol{q}, \boldsymbol{p})^{\mathsf{T}}$ and tested under the same conditions, with the initial guess for $\boldsymbol{x}^u$ generated as in Appendix E. As can be seen in Table 3, NMS is more accurate than GNODE or NODE in both the 1-D and 2-D nonlinear oscillator experiments, improving on previous results by up to two orders of magnitude. Remarkably, NMS produces more accurate entropic dynamics even in the case where the entropic variable $S$ is unobserved during NMS training and observed during the training of other methods. This illustrates another advantage of the NMS approach: because of the reasonable initial data for $S$ produced by the diffusion model, the learned metriplectic system produced by NMS remains performant even when metriplectic governing equations are unknown and only partial state information is observed.

To describe the experimental setup precisely, data is collected from a single trajectory with initial condition as $\boldsymbol{x} = (\boldsymbol{2}, \boldsymbol{0}, 0)$ following [16]. The path is calculated at 180,000 steps with a time interval of 0.001, and is then split into training/validation/test sets as before using $t_{\text{train}} = 60$, $t_{\text{val}} = 90$ and $t_{\text{test}} = 180$. Specifications of the networks used for the experiments in Table 3 are:

- **NMS:** The total number of parameters is 154. The number of layers for $\boldsymbol{A}_{\text{tri}}, \boldsymbol{B}, \boldsymbol{K}_{\text{chol}}, E, S$ is selected from {1,2,3} and the number of neurons per layer from {5,10,15}. The best hyperparameters are 1 hidden layer with 5 neurons for each network function.

- **GNODE:** The total number of model parameters is 203. The number of layers and number of neurons for each network is chosen from the same ranges as for NMS. The best hyperparameters are 1 layer with 10 neurons for each network function.

Table 3: Experimental results for the benchmark problems with respect to MSE and MAE. The best scores are in boldface.

| | 1-D D.N.O. | | T.G.C. | | 2-D D.N.O. | |
| --- | --- | --- | --- | --- | --- | --- |
| | MSE | MAE | MSE | MAE | MSE | MAE |
| **NMS**$_{\text{diff}}$ | .0170 | .1132 | .0045 | .0548 | .0275 | .1456 |
| **NMS**$_{\text{known}}$ | .0239 | .1011 | .0012 | .0276 | .0018 | .0357 |
| NODE | .0631 | .2236 | .0860 | .2551 | .0661 | .2096 |
| GNODE | .0607 | .1976 | .0071 | .0732 | .2272 | .4267 |

- NODE: The total number of model paramters is 3003. The NODE architecture is formed by stacking MLPs with Tanh activation functions. The number of blocks is chosen from {3,4,5} and the number of neurons of each MLP from {30,40,50}. The best hyperparameters are 4 and 30 for the number of blocks and number of neurons, respectively.

# D   Scaling study

To compare the scalability of the proposed NMS architecture design with existing architectures, different realizations of GNODE, GFINN, and NMS are generated by varying the dimension of the state variables, $n = \{1, 5, 10, 15, 20, 30, 50\}$. The specifications of these models (i.e., hyperparameters) are set so that the number of model parameters is kept similar between each method for smaller values of $n$. For example, for $n = 1, 5$ the number of model parameters is $\sim$20,000 for each architecture. The results in Figure 3(a) confirm that GNODE scales cubically in $n$ while both GFINN and NMS scale quadratically. Note that only a constant scaling advantage of NMS over GFINN can be seen from this plot, since $r$ is fixed during this study.

It is also worthwhile to investigate the computational timings of these three models. Considering the same realizations of the models listed above, i.e., the model instances for varying $n = \{1, 5, 10, 15, 20, 30, 50\}$, 1,000 random samples of states $\{\boldsymbol{x}^{(i)}\}_{i=1}^{1,000}$ are generated. These samples are then fed to the dynamics function $\boldsymbol{L}(\boldsymbol{x}^{(i)})\nabla E(\boldsymbol{x}^{(i)}) + \boldsymbol{M}(\boldsymbol{x}^{(i)})\nabla S(\boldsymbol{x}^{(i)})$ for $i = 1, \ldots, 1000$, and the computational wall time of the function evaluation via PyTorch's profiler API is measured. The results of this procedure are displayed in Figure 3(b). Again, it is seen that the proposed NMSs require less computational resources than GNODEs and GFINNs.

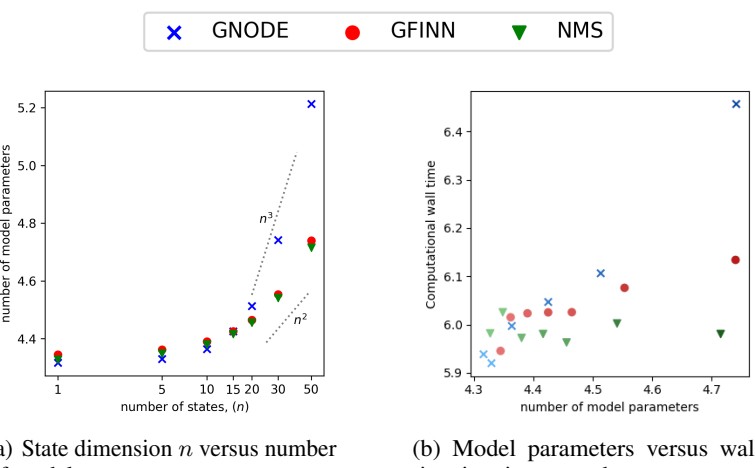

(a) State dimension $n$ versus number of model parameters

(b) Model parameters versus wall time in microseconds

Figure 3: A study of the scaling behavior of GNODE, GFINN, and NMS.

# E Diffusion model for unobserved variables

Recent work in [30] suggests the benefits of performing time-series generation using a diffusion model. This Appendix describes how this technology is used to generate initial conditions for the unobserved NMS variables in the experiments corresponding to Table 3. More precisely, we describe how to train a conditional diffusion model which generates values for unobserved variables $\boldsymbol{x}^u$ given values for the observed variables $\boldsymbol{x}^o$.

**Training and sampling:** Recall that diffusion models add noise with the following stochastic differential equation (SDE):

$$d\mathbf{x}(t) = \mathbf{f}(t, \mathbf{x}(t))dt + g(t)d\mathbf{w}, \quad t \in [0, 1],$$

where $\mathbf{w} \in \mathbb{R}^{\dim(\mathbf{x})}$ is a multi-dimensional Brownian motion, $\mathbf{f}(t, \cdot) : \mathbb{R}^{\dim(\mathbf{x})} \to \mathbb{R}^{\dim(\mathbf{x})}$ is a vector-valued drift term, and $g : [0, 1] \to \mathbb{R}$ is a scalar-valued diffusion function.

For the forward SDE, there exists a corresponding reverse SDE:

$$d\mathbf{x}(t) = [\mathbf{f}(t, \mathbf{x}(t)) - g^2(t)\nabla_{\mathbf{x}(t)}\log p(\mathbf{x}(t))]dt + g(t)d\bar{\mathbf{w}},$$

which produces samples from the initial distribution at $t = 0$. This formula suggests that if the score function, $\nabla_{\mathbf{x}(t)}\log p(\mathbf{x}(t))$, is known, then real samples from the prior distribution $p(\mathbf{x}) \sim \mathcal{N}(\mu, \sigma^2)$ can be recovered, where $\mu, \sigma$ vary depending on the forward SDE type.

In order for a model $M_\theta$ to learn the score function, it has to optimize the following loss:

$$L(\theta) = \mathbb{E}_t\{\lambda(t)\mathbb{E}_{\mathbf{x}(t)}[\|M_\theta(t, \mathbf{x}(t)) - \nabla_{\mathbf{x}(t)}\log p(\mathbf{x}(t))\|_2^2]\},$$

where $t$ is uniformly sampled over $[0, 1]$ with an appropriate weight function $\lambda(t) : [0, 1] \to \mathbb{R}$. However, using the above formula is computationally prohibitive. Thanks to [31], this loss can be substituted with the following denoising score matching loss:

$$L^*(\theta) = \mathbb{E}_t\{\lambda(t)\mathbb{E}_{\mathbf{x}(0)}\mathbb{E}_{\mathbf{x}(t)|\mathbf{x}(0)}[\|M_\theta(t, \mathbf{x}(t)) - \nabla_{\mathbf{x}(t)}\log p(\mathbf{x}(t)|\mathbf{x}(0))\|_2^2]\}.$$

Since score-based generative models use an affine drift term, the transition kernel $p(\mathbf{x}(t)|\mathbf{x}(0))$ follows a certain Gaussian distribution [32], and therefore the gradient term $\nabla_{\mathbf{x}(t)}\log p(\mathbf{x}(t)|\mathbf{x}(0))$ can be analytically calculated.

**Experimental details** On the other hand, the present goal is to generate unobserved variables $\boldsymbol{x}^u$ given values for the observed variables $\boldsymbol{x}^o = (\boldsymbol{q}, \boldsymbol{p})$, i.e., conditional generation. Therefore, our model has to learn the conditional score function, $\nabla_{\boldsymbol{x}^u(t)}\log p(\boldsymbol{x}^u(t)|\boldsymbol{x}^o)$. For example, in the damped nonlinear oscillator case, $S(t)$ is initialized as a perturbed $t \in [0, 1]$, from which the model takes the concatenation of $\boldsymbol{q}, \boldsymbol{p}, S(t)$ as inputs and learns conditional the score function $\nabla_{S(t)}\log(S(t)|\boldsymbol{q}, \boldsymbol{p})$.

For the experiments in Table 3, diffusion models are trained to generate $\boldsymbol{x}^u$ variables on three benchmark problems: the damped nonlinear oscillator, two gas containers, and thermolastic double pendulum. On each problem, representative parameters such as mass or thermal conductivity are varied, with the total number of cases denoted by $N$. Full trajectory data of length $T$ is then generated using a standard numerical integrator (e.g., dopri5), before it is evenly cut into $\lfloor T/L \rfloor$ pieces of length $L$. Let $V, U$ denote the total number of variables and the number of unobserved variables, respectively. It follows that the goal is to generate $U$ unobserved variables given $V - U$ observed ones, i.e., the objective is to generate data of shape $(NT/L, L, U)$ conditioned on data of shape $(NT/L, L, V - U)$. After the diffusion model has been trained for this task, the output data is reshaped into size $(N, T, U)$, which is used to initialize the NMS model. Note that the NODE and GNODE methods compared to NMS in Table 3 use full state information for their training, i.e., $\boldsymbol{x}^u = \varnothing$ in these cases, making it comparatively easier for these methods to learn system dynamics.

As in other diffusion models e.g. [33], a U-net architecture is used, modifying 2-D convolutions to 1-D ones and following the detailed hyperparameters described in [33]. Note the following *probability flow* ODE seen in [33]:

$$d\mathbf{x}(t) = \left[\mathbf{f}(t, \mathbf{x}(t)) - \frac{1}{2}g^2(t)\nabla_{\mathbf{x}(t)}\log p(\mathbf{x}(t))\right]dt,$$

Although models trained to mimic the probability flow ODE do not match the perofrmance of the forward SDE's result in the image domain, the authors of [30] observe that the probability flow ODE outperforms the forward SDE in the time-series domain. Therefore, the probability flow ODE is used with the default hyperparameters of [33].

