# OpenReview forum: "Efficiently Parameterized Neural Metriplectic Systems"
_NeurIPS.cc/2024/Conference — Submitted to NeurIPS 2024_

### Official Review · Reviewer_LgsG · 2024-07-12

**Soundness:** 3
**Presentation:** 2
**Contribution:** 3
**Rating:** 6
**Confidence:** 3

**Summary:**

This paper proposes a new parameterization for neural metriplectic systems which explicitly incorporates structural information about the degeneracy conditions $\{ S, \cdot \} = 0, [E,\cdot ] = 0$ into the model. The model requires $\sim O(n^2)$ learnable parameters for a problem with $n$ state variables instead of some prior methods which need $O(n^3)$. Further, it also encodes this degeneracy condition in a hard constraint, leading to models which will by construction respect these desired physical conditions. The authors provide a deep learning implementation scheme for their method which involves learning $E(x), S(x)$ and using $\nabla E, \nabla S$ to construct the matrices $L, M$ needed for the bracket from observed trajectories of the physical system. The gradients $\nabla E, \nabla S$ needed for the brackets are computed with autodifferentiation. The authors show that this system is trained end-to-end on simple physical systems including a two-gas system and a thermo-elastic pendulum and can outperform existing methods on these benchmarks.

**Strengths:**

This paper motivates the need for metriplectic systems which can be efficiently implemented and that incorporate the dynamical constraints required for these systems (energy conservation and entropy production). This captures a potentially interesting class of physical systems that could be modeled by machine learning methods like those employed in the present work. The method that they present as Algorithm 1 is straightforward and improves upon the cubic time complexity of GNODE or GFINN. The authors also provide an approximation result for their algorithm and support their claims with some experiments.

**Weaknesses:**

While the authors improve the scaling from cubic to quadratic in the number of state variables, the total complexity (quadratic) still scales poorly with size of the problem (number of state variables / dimensions). Further, the current experiments and comparisons were performed on small benchmarks. However, since this paper is the first to point out that the cubic scaling can be improved by reflecting constraints due to degeneracy, I think the experimental component of the contribution is not the most important.

**Questions:**

Lemma 3.2 involves projection matrices to construct the $L,M$ matrices. How are these chosen in the experiments?

I am not an expert in this area of ML for physical modeling. Could the authors provide some insight into the ultimate goal system they would eventually wish to model with metriplectic neural systems? Are there real world physical data which could be well captured by these kinds of models?

**Limitations:**

The authors do mention the primary limitations of this present work.

---

> ### Author Rebuttal · Authors · 2024-08-01
>
> Thank you for your thoughtful review.  Below you will find responses to your comments and questions.  Please let us know if we can do anything further to aid you in your final decision.
>
> **Weaknesses**
>
> *While the authors improve the scaling from cubic to quadratic in the number of state variables, the total complexity (quadratic) still scales poorly with size of the problem (number of state variables / dimensions). Further, the current experiments and comparisons were performed on small benchmarks. However, since this paper is the first to point out that the cubic scaling can be improved by reflecting constraints due to degeneracy, I think the experimental component of the contribution is not the most important.*
>
> We acknowledge that the experimental component of the paper is limited to small-scale problems, although it is true that the main contribution of our work is not experimental (c.f. the contributions paragraph at the end of Section 1).  We also agree that quadratic complexity in the state dimension $n$ is somewhat disappointing, but this is the minimum necessary for expressing an *arbitrary* nondegenerate metriplectic system, which can be seen based on the fact that $L$ depends on an arbitrary skew-symmetric matrix $A\in\mathbb{R}^{n\times n}$.  Future work will have to sacrifice some degree of generality to obtain a better scaling rate.
>
> **Questions**
>
> *Lemma 3.2 involves projection matrices to construct the $L,M$ matrices. How are these chosen in the experiments?*
>
> The projection matrices here are determined canonically from the gradients of the learnable energy and entropy $E,S$, and are never formed in practice (although it would be easy to do so).  Instead, the products $L\nabla E$ and $M\nabla S$ are formed directly following Lemma 3.2 and the rules of the exterior algebra.  This ensures that the necessary computations are done as compactly and efficiently as possible.
>
> *I am not an expert in this area of ML for physical modeling. Could the authors provide some insight into the ultimate goal system they would eventually wish to model with metriplectic neural systems? Are there real world physical data which could be well captured by these kinds of models?*
>
> The metriplectic formalism is especially useful for capturing dissipative perturbations of conservative systems.  An example are the Navier-Stokes equations thought of as a perturbation away from the Stokes equations.  Particularly, metriplecticity gives a useful mechanism for generating interpretable, energy-conserving and entropy-generating machine-learned models given only observed (or simulated) data.  For example, with more assumptions on the system in question (to enable better scaling), it may eventually be possible to reliably learn the metriplectic evolution of a climate system given observations at various locations around the world.  This would enable a cheap and useful surrogate for such systems which behaves in a thermodynamically realistic manner, potentially enabling better predictive capabilities.

---

> > ### Comment · Reviewer_LgsG · 2024-08-11
> >
> > I thank the authors for their rebuttal and response to my questions. I will maintain my current score.

---

### Official Review · Reviewer_dk6W · 2024-07-12

**Soundness:** 3
**Presentation:** 3
**Contribution:** 3
**Rating:** 7
**Confidence:** 3

**Summary:**

This paper proposes a parameter efficient parameterization for neural networks simulating metripletic systems, which are differential equations that have both an energy dissipative piece and an energy conservative piece. The method works by learning several of the required quantities (L and M, which trade off dissipation and conservation, I believe), while also using a small neural network to estimate the dissipation and conservation pieces (E(x) and S(x) ). As not all quantities in the state, x, can be observed, they use a time based diffusion model to emulate the hidden states (e.g. entropy) to develop initial conditions for these. Experiments are performed on two systems of this class, where it seems like the method performs better (probably due to having better inductive biases).


Unfortunately, due to not having a strong physics background, I feel somewhat unqualified to judge many of the technical strengths although things seem reasonable from a skim. I don’t know if I can properly assess novelty and significance as a result.

**Strengths:**

Significance:
-	Building better emulators of physical systems that are complicated is a good first step in what the authors term “phenomenonological” understanding of these systems.

-	Even quicker training time (and demonstrated both practically and theoretically) is quite helpful. I remember one of the original issues with NODE was that it took a very long time to converge.

Clarity:
-	Overall, the paper is pretty well written, even if quite dense, and okay to follow for a non expert physicist. I was able to follow at least the ML pieces and the experiments section quite well.

-	The relevant literature is reasonably well signposted; I learned a fair bit about the state of this field by checking the references.

Novelty:

-	The approach seems to have a clear inductive bias win over the prior works GFINN and GNODE due to better parameterization of the system.

**Weaknesses:**

Unfortunately, the writing ends up being quite dense and technical with minimal outside applications.

Sure, emulating these physical systems in the experiments is quite nice, but what types of applications does this lead to? This is more of a writing based thing and the paper could be refactored around one of these applications if possible.

**Questions:**

**Questions**

L138: Why do GFINNs require cubic scaling if they only need to learn O(rn^2) functions?

In general, what is the lower theoretical bound for learning metripletic systems?

L363: what are relevant 3-d problems of the size that you’d like to tackle?

Fig 3b: is model parameters measured in thousands here? Don’t think we can have 4.3 parameters…

Could the authors generate plots like Fig 2 for the other two experiments? I think it may be useful to understand why the double pendulum example shows less relative improvement in that system?

Are the two main text experiments fully observed? Could the authors run these without full observations (e.g. with the diffusion for the unobserved states) if that makes sense in this setting?

L226: why tanh activation for this task?

**Writing comments:**

-	I would personally suggest moving a bit more of appendix E into the main text, as well as Fig 3. These are both interesting and pretty compelling.

-	I would also suggest then moving section 3.1 into the appendix as I don’t think you really need much of that material. Pieces of Section 2 are much too dense and could be moved into the appendix as well; it should probably be half a page at most.

---

> ### Author Rebuttal · Authors · 2024-08-01
>
> Thank you for your thoughtful review. We are glad to hear that you found our paper well-written and could use it to study the current  literature in this interesting area of ML.
>
> Below you will find responses to your comments and questions.  Please let us know if we can do anything further to aid you in your final decision.
>
> **Weaknesses**
>
> *Unfortunately, the writing ends up being quite dense and technical with minimal outside applications.*
>
> *Sure, emulating these physical systems in the experiments is quite nice, but what types of applications does this lead to? This is more of a writing based thing and the paper could be refactored around one of these applications if possible.*
>
> The primary application of this work lies in producing more interpretable and physically realistic models from data.  Because the models produced by our NMS method are provably metriplectic, the conservative and dissipative contributions to the learned dynamics can be clearly identified, the dynamical evolution is (provably) stable, and the generalization error in the model is (provably) bounded.  This contributes to the utility of the metriplectic formalism in phenomenological modeling, since any observed or simulated data can be fit to a metriplectic system in a way that "energy" is conserved and dissipative effects are captured through the generation of "entropy".  This allows for more physically realistic surrogate models which are seen to train easier and generalize better than structure-uninformed approaches like NODE.
>
> **Questions**
>
> *L138: Why do GFINNs require cubic scaling if they only need to learn O(rn^2) functions?*
>
> This is because the $r$ could be almost as large as $n$.  More precisely, we do not know the rank $r$ of the dissipation matrix $M$ ahead of time, except that $1\leq r\leq n-1$.  So, the scaling is potentially cubic in $n$.
>
> *In general, what is the lower theoretical bound for learning metripletic systems?*
>
> As illustrated by our parameterizations in Lemma 3.2, the lower bound on learnable functions for a *general* nondegenerate metriplectic system is quadratic in both $n$ and $r$.  This is because $L$ depends on a general skew-symmetric matrix field $A$ described by $n(n-1)/2$ functions and similarly $M$ depends on a symmetric Cholesky factor containing $r(r+1)/2$ functions.  To do better than this, we must give up on full generality and assume more knowledge about the metriplectic system in question, which is an interesting avenue for future work.
>
> *L363: what are relevant 3-d problems of the size that you’d like to tackle?*
>
> It would be highly interesting to apply an NMS-like approach to large-scale fluid problems such as the Navier-Stokes equations, or to tackle observational data about the climate or atmosphere.  On the other hand, the method in this paper cannot apply directly to these scenarios due to its quadratic scaling in the state dimension $n$, and so more than just metriplectic structure will have to be assumed in order to improve this rate.
>
> *Fig 3b: is model parameters measured in thousands here? Don’t think we can have 4.3 parameters…*
>
> Yes, thanks for the catch.  We have corrected this in the revised manuscript.
>
> *Could the authors generate plots like Fig 2 for the other two experiments? I think it may be useful to understand why the double pendulum example shows less relative improvement in that system?*
>
> We have done this, and included the relevant plots in the Appendix to the revised manuscript.  We observe that the double pendulum is more difficult to train for all methods due to its inherent complexity, which is likely why there is less relative improvement in this case. Please refer to the PDF file in the global response.
>
> *Are the two main text experiments fully observed? Could the authors run these without full observations (e.g. with the diffusion for the unobserved states) if that makes sense in this setting?*
>
> Good question.  The main experiments are *not* fully observed, which in particular makes them "harder": the network has to figure out the evolution of entropy density on its own.  Assuming access to the full state data improves results substantially, as can be seen in Table 3 in Appendix C.
>
> *L226: why tanh activation for this task?*
>
> We choose tanh because of its smoothness.  Our theoretical results assume a sufficient degree of regularity in $L,M,E,S$ (at least $C^1$ in most cases), and tanh ensures that this holds in our architectures.  On the other hand, many different activations could be used instead, and it is possible (maybe even likely) that infinite regularity is not the best option in all cases.
>
> **Writing comments**
>
> - *I would personally suggest moving a bit more of appendix E into the main text, as well as Fig 3. These are both interesting and pretty compelling.*
> - *I would also suggest then moving section 3.1 into the appendix as I don’t think you really need much of that material. Pieces of Section 2 are much too dense and could be moved into the appendix as well; it should probably be half a page at most.*
>
> Thank you for the suggestions.  In view of your comments, in the revised version we have moved some of the details in Section 2 into the Appendix in order to move some more of Appendix E (plus Fig 3) into the main text.  We have elected to keep Section 3.1 as it is, since this material is essential for understanding the NMS method.

---

> ### Comment · Reviewer_dk6W · 2024-08-11
>
> Thanks for responding to my questions. I now have a bit better understanding of this paper, and still think it should be accepted.
>
> > L138
>
> Thanks, that makes sense, although I believe it could reduce the computational gains compared to the other methods.
>
> Thank you for also attaching the figures for double pendulum. I see that exploiting the structure as your approach does tends to improve the qualitative performance as well.

---

### Official Review · Reviewer_Lujy · 2024-07-13

**Soundness:** 3
**Presentation:** 3
**Contribution:** 2
**Rating:** 5
**Confidence:** 1

**Summary:**

This work presents a method for learning metriplectic systems from data. Metriplectic systems are a model which conserve energy and produce entropy, two desirable features. Their method, termed “neural metriplectic systems” (NMS), is based on a more efficient system parametrization. The authors also prove universal approximation results on non-degenerate systems, and generalization error bounds. They verify that their method outperforms other metriplectic-learning baselines, GNODE and GFINN, on two physical experiments.

**Strengths:**

Originality: Although I am not at all an expert in this field and therefore cannot properly judge, it seems that the main theorem (Theorem 3.4) is novel and non-trivial.

Quality: The proposed method outperforms baselines in two experimental settings, verifying the expected gain from having a more efficient parametrization. The corresponding theoretical results, on universality and generalization, provide a fairly thorough picture of the method.

Significance: Within the field of learning metriplectic systems, this paper seems to make a valuable contribution and improve on prior work.

Clarity: The paper is very well-written, and mathematically rigorous. Although the details are not accessible to someone without a background that matches the subject material rather closely, the high-level ideas about the benefits of metriplectic systems, what past work has done, and the advantages of their new method, are conveyed well.

**Weaknesses:**

Clarity:
1. Although well-written, the paper is not accessible to most machine learning audiences, and seemingly requires the reader to already have a physics background in phenomenological modeling, or exterior algebra.

2. Mathematical terms such as algebraic brackets, Poisson brackets, degenerate metric brackets, etc. should be defined in the beginning of the paper, or with a reference to a textbook or other paper defining them. The “exterior algebra” background is suitable for only those with a strong mathematical background already, using terms like “wedge product” and “graded algebra” without definition. (Admittedly, it would be impossible to fully explain all of these concepts in only 9 pages — perhaps a citation to a textbook would be helpful here, but in practice if the reader needs to understand the decomposition result properly to grasp the contribution, then this work may be more suitable for a venue other than a machine learning conference.)

Quality: The baselines in experiments, as well as the methods discussed in the exposition, are all metriplectic. However, it seems like other methods (e.g. which preserve energy but do not increase entropy, or even those which are not physics-informed at all), should be included too.

Significance: I am not sure how widely applicable metriplectic learning systems are, or what alternative (non-metriplectic) methods can be used for the same problems. The paper would be improved by providing more of this background/motivation.

Overall, as a non-expert, my main concern is with the suitability of this work for a machine learning conference - I defer to the AC on this point. It seems that the machine learning techniques used within NMS are fairly straightforward, while it is the parametrization in Theorem 3.4 that seemingly constitutes the crux of the method. However, the statement and proof of Theorem 3.4 would be more accessible to a physics or math audience, than an ML audience.

**Questions:**

1. NMS is a more efficient parametrization than GNODE and GFINN. Is it also more efficient computationally (end-to-end)? (For example, one could in theory imagine a system with very few parameters, that uses them in a forward pass in a computationally intense way.)

2. Are there non-metriplectic methods that are suitable for comparison in the experiments (for example, methods that preserve energy like Hamiltonian networks)?

3. What is the motivation for metriplectic methods overall — do they have drawbacks relative to methods that do not preserve energy/increase entropy? It would be helpful to include these in related work.

**Limitations:**

Limitations are discussed.

---

> ### Author Rebuttal · Authors · 2024-08-01
>
> Thank you for your thoughtful review.  We are glad to hear that you found our paper well-written and reflective of a valuable contribution.
>
> Below you will find responses to your comments/questions, including your main concern that the work may not be well-suited for a machine learning audience.  For space requirements, we have occasionally truncated your original (italicized) comments with "...". Please let us know if we can do anything further to aid you in your final decision.
>
> *Overall, as a non-expert, my main concern is with the suitability of this work for a machine learning conference - I defer to the AC on this point. It seems that the machine learning techniques used within NMS are fairly straightforward, while it is the parametrization in Theorem 3.4 that seemingly constitutes the crux of the method. However, the statement and proof of Theorem 3.4 would be more accessible to a physics or math audience, than an ML audience.*
>
> It is true that the parameterizations in Theorem 3.4 and the associated theoretical results (Proposition 3.7 and Theorem 3.9) are a primary contribution of this work.  However, NMS is fundamentally a machine learning method: the goal is to learn a nondegenerate metriplectic system from data.  We think that the use of mathematically rigorous machinery in pursuit of this goal enhances the value of the work and does not disqualify us from high-quality machine learning venues such as NeurIPS.  Indeed, the presented mathematics have allowed us to decouple metriplectic structure-preservation from any optimization error incurred during training, leading to a provably structure-preserving model while simultaneously guaranteeing universal approximation and meaningful estimates of the error in the trajectories.  Note that several related works have been published in NeurIPS.  For example:
>
> - Greydanus et al. Hamiltonian neural networks. NeurIPS 2019
> - Finzi et al. Simplifying Hamiltonian and Lagrangian Neural Networks via Explicit Constraints. NeurIPS 2020
> - Chen et al. Neural symplectic form: Learning Hamiltonian equations on general coordinate systems. NeurIPS 2021
> - Lee et al. Machine learning structure preserving brackets for forecasting irreversible processes. NeurIPS 2021
> - Gruber et al. Reversible and irreversible bracket-based dynamics for deep graph neural networks. NeurIPS 2023
>
> *Quality: ...it seems like other methods (e.g. which preserve energy but do not increase entropy, or even those which are not physics-informed at all), should be included too.*
>
> We agree that additional experiments are always helpful.  Note that the case of networks which are not physics-informed has already been handled by the NODE architecture (see Tables in Section 5), and it is clear that this approach is not as performant as NMS.  The case of purely energy-conserving networks (e.g., Hamiltonian NNs) is being included in the revision, and is also notably less performant than NMS.  This is expected, since the underlying dynamics are not Hamiltonian.
>
> *Significance: I am not sure how widely applicable metriplectic learning systems are...*
>
> The metriplectic formalism has been widely adopted in nonequilibrium thermodynamics, and a large variety of interesting physical systems can be understood through this lens.  Besides the references mentioned in the Introduction, a representative monograph with many examples is [1].  While we are limited by space requirements, we have added a bit more background information to the revised Section 1.
>
> [1] Öttinger, Hans Christian. Beyond equilibrium thermodynamics. Vol. 5. Hoboken: Wiley-Interscience, 2005.
>
> **Weaknesses**
>
> *1.	Although well-written, the paper is not accessible to most machine learning audiences...*
>
> It is true that a background in the relevant mathematics is useful for fully understanding our work.  However, we do not assume that all readers possess this knowledge; references are left for those seeking more background, and the paper has been written so that non-experts can follow the main ideas at a high level.
>
> *2.	 Mathematical terms such as algebraic brackets, Poisson brackets, degenerate metric brackets, etc. should be defined in the beginning of the paper, or with a reference to a textbook or other paper defining them...*
>
> Note that the first set of requested terms are defined in the second paragraph of page 2, and the second in Section 3.1, albeit in a terse way because of the space requirements.  Since this was not immediately clear, we have attempted to include a few more details and citations in the revised manuscript.
>
> **Questions**
>
> *1.	NMS is a more efficient parametrization than GNODE and GFINN. Is it also more efficient computationally (end-to-end)?*
>
> This is a good question.  While NMS is not epoch-to-epoch competitive with the structure-agnostic NODE in cost, we observe that it does perform better than GNODE and GFINN for the same number of parameters.  This is illustrated in Figure 3b in Appendix D.
>
> *2.	Are there non-metriplectic methods that are suitable for comparison in the experiments?*
>
> As mentioned above, we plan to include a comparison to Hamiltonian NNs in the revised manuscript.
>
> *3.	What is the motivation for metriplectic methods overall — do they have drawbacks relative to methods that do not preserve energy/increase entropy? It would be helpful to include these in related work.*
>
> Metriplectic methods are designed to encode the first two laws of thermodynamics in a way that is suitable for modeling physical systems with dissipation.  By capturing dissipative phenomena through entropy gain, metriplectic models are interpretable and thermodynamically closed.  Their primary drawback is theoretical: it is difficult to write physical systems in metriplectic form, and there is no general algorithm for doing so.  This is what motivates machine learning approaches like NMS, which remove the need for this difficult step.  For clarity, we have added a few more details about this in the revised manucript.

---

> > ### Comment · Reviewer_Lujy · 2024-08-12
> > **Thanks for the response**
> >
> > Thanks to the authors for their response. I retain my low confidence rating and concern about the understandability of the paper, but will increase my score.

---

### Author Rebuttal · Authors · 2024-08-06

Thanks to the reviewers for their helpful feedback and interest in our article.  We are pleased to hear that all reviewers consider it a valuable contribution to the field of structure-preserving machine learning.

In response to reviewer dk6W's comment, we are attaching a set of figures for the double pendulum example.  Here, we show the trajectories of the ground-truth and predicted state variables, entropy, and energy. Note that the proposed NMS method produces more accurate predictions than previous methods while preserving the metriplectic structure of the system.

Additionally, individualized responses to each review are left below.  Please let us know if we can do anything further to aid you in your final decision.

---

### Decision · Program_Chairs · 2024-09-25

**Decision:**

Reject

**Comment:**

This paper introduces neural metriplectic systems (NMS) for learning metriplectic systems from data. The proposed approach offers efficient parametrization, with its performance demonstrated on two physical experiments. The approach is novel and sound, with the authors providing universal approximation results on non-degenerate systems and generalization error bounds. However, the experimental results are somewhat weak, and the small improvement in model performance, in terms of mean squared error on the thermoelastic double pendulum example, appears practically insignificant. Including more compelling real-world applications would have better showcased the model's capabilities. Additionally, the scalability of the approach seems irrelevant in practice, as the models used are relatively "tiny."

Given these points, I am not entirely convinced of the proposed approach’s practical relevance, and I have doubts about its widespread adoption. While the developed theory and insights are stimulating, it remains unclear how they will inspire further downstream work. The conclusion mentions future work extending NMS to infinite-dimensional metriplectic systems to address its main limitation: difficulty scaling to realistic 3D problems. I strongly suggest including such experiments in this paper to strengthen its practical impact.

While the reviews are positive, the reviewers express limited confidence. I feel that the weaknesses outweigh the strengths of this paper, and thus I recommend rejecting it.